# Dereplication of microbial metabolites through database search of mass spectra

Hosein Mohimani[1,2], Alexey Gurevich[3], Alexander Shlemov [3], Alla Mikheenko[3], Anton Korobeynikov [3,4], Liu Cao[1], Egor Shcherbin[5], Louis-Felix Nothias[6], Pieter C. Dorrestein[6,7] & Pavel A. Pevzner [2,3]

Natural products have traditionally been rich sources for drug discovery. In order to clear the road toward the discovery of unknown natural products, biologists need dereplication strategies that identify known ones. Here we report DEREPLICATOR+, an algorithm that improves on the previous approaches for identifying peptidic natural products, and extends them for identification of polyketides, terpenes, benzenoids, alkaloids, flavonoids, and other classes of natural products. We show that DEREPLICATOR+ can search all spectra in the recently launched Global Natural Products Social molecular network and identify an order of magnitude more natural products than previous dereplication efforts. We further demonstrate that DEREPLICATOR+ enables cross-validation of genome-mining and peptidogenomics/glycogenomics results.

[1] Computational Biology Department, School of Computer Sciences, Carnegie Mellon University, Pittsburgh, PA, USA. [2] Department of Computer Science and Engineering, University of California, San Diego, La Jolla, CA, USA. [3] Center for Algorithmic Biotechnology, Institute of Translational Biomedicine, St. Petersburg State University, St. Petersburg, Russia. [4] Department of Statistical Modelling, St. Petersburg State University, St. Petersburg, Russia. [5] National Research University Higher School of Economics, St. Petersburg, Russia. [6] Collaborative Mass Spectrometry Innovation Center, Skaggs School of Pharmacy and Pharmaceutical Sciences, University of California, San Diego, La Jolla, CA, USA. [7] Department of Pharmacology and Pediatrics, University of California, San Diego, La Jolla, CA, USA. Correspondence and requests for materials should be addressed to H.M. (email: hoseinm@andrew.cmu.edu)

Since 1990s, there has been a decline in the pace of anti-biotics discovery from natural sources[1]. However, natural product discovery has recently gained attraction due to multiple technological advances, exemplified by the discovery of teixobactin[2,3]. Global Natural Products Social (GNPS) molecular networking project[4] is a recent mass spectrometry data repository in the field of natural products. While thousands of laboratories have contributed billion mass spectra, identifying spectra of known natural products in this repository is a challenging problem. The first attempts to solve this problem date back to 1960s[5–8], two decades before mass spectrometry-based database search tools appeared in proteomics[9,10]. However, to date, computational mass spectrometry of small molecules is regarded as a less mature field than the proteomic counterpart[11–18].

One of the main challenges in the field of natural product is the high rate of re-discovery of known natural products. The process of using the information about the chemical structure of a known natural product to identify this compound in an experimental sample (without having to repeat the entire isolation and structure-determination process) is called dereplication. Development of chemical structure databases such as PubChem (≈83 million compounds), ChemSpider (≈58 million compounds), ChEMBL (2.1 million compounds), ChemBank (1.2 million compounds), ChEBI (440,000 compounds), Dictionary of Natural Products (≈300 thousand compounds), AntiMarin (≈60 thousand compounds), KEGG (≈16 thousand compounds), MetaCy (≈10 thousand compounds), mzCloud (≈3 thousand compounds), NuBBEDB (2500 compounds), MIBiG (≈1600 compounds), DrugBank (1360 compounds with structural information), and Norine (≈1000 compounds) has paved the way for development of bioinformatics tools for natural product dereplication. Early dereplication approaches were based on deriving the exact chemical formula using high-resolution precursor mass and searching for compounds described by this formula in chemical structure databases[19–22]. However, this approach often fails because the number of possible formulas rapidly increases with the molecular mass of metabolites and because existing chemical databases contain many compounds with identical formulas.

Several strategies have been proposed for dereplicating spectra of metabolites, including (i) combinatorial fragmentation strategies that use systematic bond disconnection approach to break bonds between heavy atoms[23–28], (ii) the HighChem Mass Frontier approach that predicts fragmentation based on standard reactions and a library of fragmentation rules; (iii) approaches based on learning the mapping between mass spectra and molecular formula of the candidate compounds from reference spectra to create predicted fragmentation trees and possible structures against which mass spectra can be searched[29,30]; (iii) approaches that use stochastic Markov modeling for simulating mass spectra from molecular structures and matching them against experimental spectra[31]; (iv) approaches that construct in silico mass spectra by fragmenting peptides and lipids along specific bonds[32,33]; (v) approaches that annotate the structural motifs rather than the entire structure[34]; (vi) the ab initio approaches that predict the likely fragmentation of a molecule by computing its energetic landscape[35].

Currently, the fast spectral library search programs[36] search over 1000 spectra against the entire NIST library per seconds. However these approaches are unable to directly search chemical structure libraries. Despite recent progress (CSI:FingerID[29] increased metabolite identification rates fivefold as compared to previous approaches), the existing tools for metabolite identifications are either limited to specific classes of molecules such as peptides and lipids[32,33], work best for identification of small molecules (below 500 Da)[29], or become prohibitively time-consuming for searching large spectral datasets[29]. Recently, we

introduced DEREPLICATOR[32] for searching spectral datasets against the database of peptidic natural products (PNPs) that include nonribosomal peptides (NRPs), and ribosomally synthesized and post-translationally modified peptides (RiPPs). DEREPLICATOR constructs theoretical spectra of peptide natural products by disconnecting all the bridges and 2-cuts representing amide bonds, and measuring the masses of the connected components. By applying spectral networks[37], DEREPLICATOR enabled identification of variants of known PNPs. While DEREPLICATOR search of GNPS identified hundreds of peptides and their variants, it is limited to dereplicating PNPs and cannot identify other classes of natural products such as polyketides and terpenes.

Here we describe DEREPLICATOR+ algorithm for dereplicating spectra against diverse metabolites. After searching nearly two hundred million tandem mass spectra in the GNPS molecular networking infrastructure, DEREPLICATOR+ identifies five times more molecules than the previous approaches. DEREPLICATOR+ enables high-throughput identification of variants of known natural products by spectral networks.

## Results

**Outline of the DEREPLICATOR+ algorithm.** Figure 1 and Supplementary Figure 1 show the DEREPLICATOR+ pipeline that includes the following steps described in the Methods section: (i) constructing metabolite graphs from metabolite chemical structures, (ii) generating fragmentation graphs, (iii) constructing decoy fragmentation graphs, (iv) annotating target and decoy fragmentation graphs by spectra and scoring metabolite-spectrum matches (MSMs), (v) computing statistical significance of MSMs and evaluating the false discovery rate (FDR), and (vi) enlarging the set of MSMs by molecular networking (Fig. 1).

**Datasets.** To benchmark DEREPLICATOR+, we used the AntiMarin database (60,908 compounds and 29,491 unique compounds) and the Dictionary of Natural Products (254,727 compounds and 83,889 unique compounds). Compounds are flagged as duplicates if they have identical chemical structures. We searched all spectra from the following reversed-phase liquid chromatography high-resolution mass spectrometry datasets specified in Supplementary Table 1 and Supplementary Note 1. Spectra$_{ActiSeq}$ (178,635 spectra from MSV000078604 and 473,135 spectra from MSV000078839) contains spectra obtained from bacterial extracts of 36 strains of *Actinomyces* with published draft genomes. The spectral dataset is partitioned into 36 subsets corresponding to these strains[32,38]. Spectra$_{Library}$ (5473 annotated spectra) contains a combination of spectral libraries from the GNPS public library, the NIH natural products library, the Food and Drug Administration natural products library, MASSBANK, HMDB, and RESPECT, all available from GNPS. Spectra$_{Lichen}$ (926,864 spectra from MSV000078584) contains spectra obtained from extracts of *Peltigera* sp[39]. Illumina paired end reads (82,722,940) of length 250 bp obtained from the same sample were assembled by metaSPAdes[40]. The total length of contigs longer than 1000 bp in the assembled metagenome is 136 Mb. Spectra$_{Cyan}$ (11,921,457 spectra from MSV000078568) contains spectra from extracts of cyanobacterial strains, partitioned into 317 datasets corresponding to individual collections[41]. Four of these strains have their genomes available (*Moorea bouillonii* PNG19MAY05, *Moorea producens* JHB22AUG96, *M. producens* NAK12DEC93-3La, and *M. producens* PAL15AUG08). Spectra$_{GNPS}$ (248.1 million spectra) contains spectra from 555 GNPS datasets with 77,045 samples (deposited by over 200 labs before August 2017). Spectra$_{Fungi}$, Spectra$_{Acti}$, and Spectra$_{Pseudo}$ refer to subsets of Spectra$_{GNPS}$ containing spectra from Fungi,

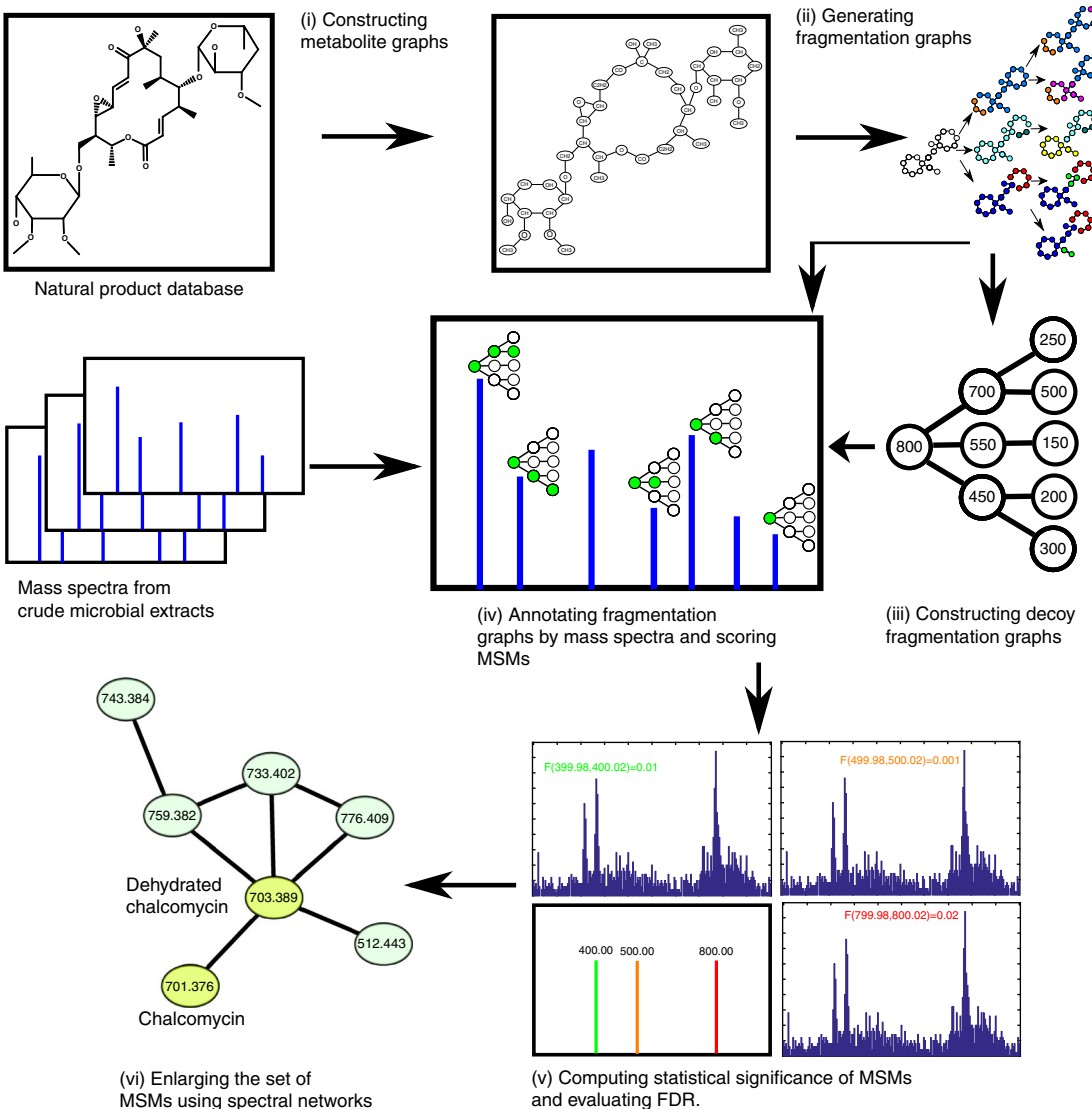

**Fig. 1** DEREPLICATOR+ pipeline. The method consists of (i) constructing metabolite graphs from metabolites chemical structures, (ii) generating fragmentation graphs, (iii) constructing decoy fragmentation graph for each metabolite (iv) annotating target and decoy fragmentation graphs by spectra and scoring metabolite-spectrum matches (MSMs), (v) computing statistical significance of MSMs and evaluating FDR, and (vi) enlarging the set of MSMs by molecular networking

*Actinomyces*, and *Pseudomonas*, respectively. To identify the spectra coming from media and contamination, several analyzed samples (called blank samples) consisted only of media with no bacteria cultured in them.

**Discovering chalcomycin variants from *Actinomyces* spectra.** At 1% FDR (corresponding to the $p$ value threshold $10^{-7}$ computed by MS-DPR[42]), DEREPLICATOR identified 73 unique compounds (166 MSMs) in Spectra$_\text{ActiSeq}$. At 0% FDR (corresponding to the $p$ value threshold $10^{-8}$), DEREPLICATOR identified 66 unique compounds (148 MSMs). In contrast, at 1% FDR (score threshold of 6), DEREPLICATOR+ identified 488 compounds and 8194 MSMs (1.2% of all spectra in Spectra$_\text{ActiSeq}$ dataset). At 0% FDR (score threshold of 9), it identified 154 compounds (2666 MSMs), a twofold increase as compared to DEREPLICATOR.

DEREPLICATOR+ not only identified more unique compounds at the same FDR, but also identified more spectra per compounds (average of 2.2 spectra per compound for DEREPLICATOR, versus 16.7 spectra per compound for DEREPLICATOR+). This is partially because spectra from the same compound often differ in the quality of fragmentation. DEREPLICATOR is mainly limited to identification of the highest quality spectra since it uses a rather restrictive fragmentation model. DEREPLICATOR+ identifies spectra of lower quality since it uses a more detailed fragmentation model.

Among 154 compounds identified by DEREPLICATOR+ at 0% FDR, ClassyFire[43] classified 92 of them as peptides and amino acids derivatives, 32 as lipids, 5 as benzenoids, and 6 as other classes. Out of 154, 72 of these compounds have the known *Actinomyces* origin according to AntiMarin (Supplementary Data 1).

To analyze some of DEREPLICATOR+ identifications in more detail, we selected a very stringent score threshold of 15 (0% FDR, 29 compounds) and removed four compounds that were present in blank samples from the background media. Table 1 describes $24 = 29 - 5$ compounds (covering 19 PNPs, 2 polyketides (PKs), 2 terpenes, and 1 benzenoid) identified by DEREPLICATOR+. These 24 metabolites form 15 metabolite families and reveal additional 557 variants of these metabolites through molecular

**Table 1 The list of top scoring 25 metabolites identified by DEREPLICATOR+in the search of the Spectra$_{ActiSeq}$ dataset against the AntiMarin database at the score threshold of 15**

| compound | class | CN | CO | DEREP+ p-value | DEREP p-value | producer | ref | gene similarity | Cmp |
|---|---|---|---|---|---|---|---|---|---|
| stenothricin-III | peptide | 21 | 4 | $9.10^{-22}$ | $9.10^{-14}$ | S. roseosporous | [63] | BGC0000431(100%) | 0 |
| doricin | peptide | 16 | 4 | $3.10^{-27}$ | $2.10^{-10}$ | S. pristinaespiralis | [64] | BGC0000952(97%) | 74 |
| arylomycin-A2 | peptide | 14 | 5 | $5.10^{-15}$ | $4.10^{-7}$ | S. roseosporous | [65] | BGC0000306(88%) | 26 |
| WS-9326-A | peptide | 16 | 5 | $1.10^{-20}$ | $4.10^{-17}$ | S. griesoflavus | [66] | BGC0001297(100%) | 2 |
| arylomycin-A4 | peptide | 14 | 5 | $3.10^{-15}$ | $2.10^{-9}$ | S.roseosporous | [65] | BGC0000306(88%) | 26 |
| ostreogrycin-B | peptide | 16 | 3 | $1.10^{-17}$ | $7.10^{-8}$ | S. pristinaespiralis | [64] | BGC0000952(97%) | 74 |
| SP-Chymostatin-B | peptide | 12 | 2 | $6.10^{-9}$ | $4.10^{-10}$ | Streptomyces sp. E14 | [67] | unknown | 95 |
| pristinamycin-IC | peptide | 16 | 3 | $1.10^{-15}$ | $2.10^{-9}$ | S. pristinaespiralis | [64] | BGC0000952(97%) | 0 |
| salinamide-E | peptide | 13 | 7 | $9.10^{-13}$ | 0.002 | Streptomyces CNH287 | [68] | BGC0001230(100%) | 0 |
| antimycin-B1 | benzenoid | 5 | 8 | $3.10^{-7}$ | $5.10^{-6}$ | Streptomyces albus | [62] | BGC0000958(86%) | 15 |
| virginiamycin-S1 | peptide | 13 | 3 | $2.10^{-23}$ | $9.10^{-15}$ | S. pristinaespiralis | [64] | BGC0000952(97%) | 74 |
| ostreogrycin-A | peptide | 4 | 4 | $2.10^{-17}$ | 0.002 | S. pristinaespiralis | [64] | BGC0000952(97%) | 6 |
| actinomycin-X2 | peptide | 24 | 4 | $5.10^{-19}$ | $3.10^{-10}$ | Streptomyces CNS654 | [69] | BGC0000296(71%) | 2 |
| A-21978-C2 | peptide | 30 | 7 | $8.10^{-13}$ | $3.10^{-9}$ | S. roseosporous | [70] | BGC0000952(59%) | 0 |
| soyasaponin-I | triterpene | 1 | 21 | $1.10^{-12}$ | 1 | S. hygroscopicus | [71] | unknown | 53 |
| C35H56O13 | polyketide | 2 | 18 | $3.10^{-10}$ | 1 | S. Mg1 | [46] | unknown | 4 |
| nocardamine | peptide | 12 | 1 | $4.10^{-15}$ | $3.10^{-8}$ | S. Mg1 | [46] | unknown | 2 |
| ostreogrycin-G | peptide | 4 | 4 | $3.10^{-22}$ | 0.001 | S. pristinaespiralis | [49] | BGC0000952(97%) | 111 |
| virginiamycin-M1A | peptide | 4 | 4 | $6.10^{-17}$ | 0.02 | S. pristinaespiralis | [49] | unknown | 0 |
| virginiamycin-S2 | peptide | 13 | 4 | $6.10^{-21}$ | $4.10^{-10}$ | S. pristinaespiralis | [49] | BGC0000952(97%) | 7 |
| salinamide-A | peptide | 14 | 8 | $1.10^{-19}$ | $1.10^{-7}$ | Streptomyces CNB091 | [68] | BGC0001230(100%) | 5 |
| chalcomycin | polyketide | 1 | 18 | $1.10^{-9}$ | 1 | S. Mg1 | [46] | BGC0000047(64%) | 3 |
| soyasaponin-II | triterpene | 1 | 20 | $2.10^{-10}$ | 1 | S. Tu6071 | [71] | unknown | 53 |
| WA-3854-A2 | peptide | 4 | 5 | $1.10^{-8}$ | $6.10^{-7}$ | S. ghanaensis | [72] | unknown | 9 |

For each compound we show its classification by ClassyFire (class), a software tool for metabolite classification[43], as well as the DEREPLICATOR+ p-value (DEREP+ p-value), and the DEREPLICATOR p-value (DEREP p-value). DEREPLICATOR p-values are computed using MS-DPR method[42]. For DEREPLICATOR+, p-value computation is described in the METHOD section. In all cases, the compounds have been reported in another *Actinomyces* species, and the corresponding references are shown. In 17 out of 24 cases, the compounds have known BGC, and in all these cases DEREPLICATOR+ identifications were validated by the BLAST search of the BGC. In each case, the number of compounds in the connected components of the molecular network (Cmp) for each identified metabolite is also shown. While soyasaponin was first discovered in plants[73], it was shown later that it is also produced by *Streptomyces*[71]. Number of nitrogen to carbon bonds (CN), and oxygen to carbon bonds (CO) in the molecular structures are also shown. Supplementary Data 1 is an extended version of this table, including a comprehensive list of all the 488 identifications of DEREPLICATOR+ in Spectra$_{Acti}$ at 1% FDR

networks[37,44]. DEREPLICATOR missed 10 out of these 24 metabolites at 3% FDR (2 PKs, 2 terpenes, 1 benzenoid, and 5 short PNPs with <8 amide bonds). Supplementary Data 1 describes all the 488 identifications of DEREPLICATOR+ in *Actinomyces* spectra at 1% FDR threshold.

In 17 out of 24 cases, the compounds had a known biosynthetic gene cluster (BGC) reported in the MIBiG database, and in all these cases we cross-validated DEREPLICATOR+ identifications by searching their BGCs against the known genome of their producers (BLAST search found very similar homologs). Although the BGC is currently unknown for antibiotic $C_{35}H_{56}O_{13}$ (originally isolated from *Streptomyces hirsutus*)[45], we used $C_{35}H_{56}O_{13}$ identification in *Streptomyces* Mg1 to derive its BGC. Analysis of the PK gene clusters reported by antiSMASH in the *Streptomyces* Mg1 genome led to assignment of antibiotic $C_{35}H_{56}O_{13}$ to a PK BGC in *Streptomyces* Mg1. Kersten et al.[46] showed that this BGC is responsible for production of PK chalcomycin. The antibiotic $C_{35}H_{56}O_{13}$ is very similar to chalcomycin (differing only in a single monomer methylation), implying that this BGC is likely responsible for the production of both chalcomycin and antibiotic $C_{35}H_{56}O_{13}$. Analysis of the number of peptide bonds in the compounds identified by DEREPLICATOR and DEREPLICATOR+ revealed that DEREPLICATOR+ outperforms DEREPLICATOR in the identification of compounds with a small number of peptide bonds (Fig. 2).

DEREPLICATOR+ analysis of the Spectra$_{ActiSeq}$ dataset identified 56 out of 207 compounds in the DNP annotated as Actionmyces/Streptomyces (Supplementary Data 2). Supplementary Figure 2 shows a comparison between DEREPLICATOR p values and DEREPLICATOR+ p values for 24 top identifications in Table 1.

After running DEREPLICATOR, DEREPLICATOR-CN, DEREPLICATOR-CN-CO, and DEREPLICATOR-G (defined in Methods sections) on Spectra$_{ActiSeq}$ with FDR threshold 3%, they identified 75, 91, 404, and 496 compounds, respectively. In contrast, DEREPLICATOR+ identifies 1024 compounds at this FDR. DEREPLICATOR+ significantly improves on DEREPLICATOR in the case of short peptides with small number of amide bonds. As an example, DEREPLICATOR fails to identify the antibiotic arylomycin A4, a branch-cyclic peptide with five amino acids, at 3% FDR, while DEREPLICATOR+ discovered it at 0% FDR with p value $3 \times 10^{-15}$. Supplementary Figure 3 compares arylomycin A4 annotation of DEREPLICATOR and DEREPLICATOR+.

Supplementary Note 2, Supplementary Data 1, 3, and 4, and Supplementary Figure 4 describe the performance of DEREPLICATOR+ on different fragmentation methods (collision-induced dissociation (CID) and higher-energy C-trap dissociation (HCD)), identification of frequent mass spectrometry contaminants[47], and results of searches of the larger databases.

The NIST spectral library search tool MSPepSearch identified 34, 12, and 12 compounds in searching Spectra$_{ActiSeq}$ against NIST17, LipidBlast and MoNA. At 1% FDR DEREPLICATOR+ recovered 27/34 identifications from NIST, 8/12 identifications from LipidBlast, and 9/12 identifications from MoNA (Supplementary Data 5). At this FDR threshold DEREPLICATOR+ identified 315 compounds, 272 of them absent from LipidBlast, MoNA, and NIST search results.

**Benchmarking DEREPLICATOR+ on spectral library.** To benchmark the accuracy of DEREPLICATOR+ in identification of spectra from known compounds, we searched 5473 annotated spectra from Spectra$_{Library}$ against a database of their 5473 chemical structures, plus 83,889 distinct chemical structures from the DNP database. We removed 2697 duplicate compounds that were shared between the spectral library and the DNP database, resulting in 86,665 distinct compounds. By allowing a large precursor mass tolerance of 0.5 Da, each spectrum is searched against 1235 compounds on average.

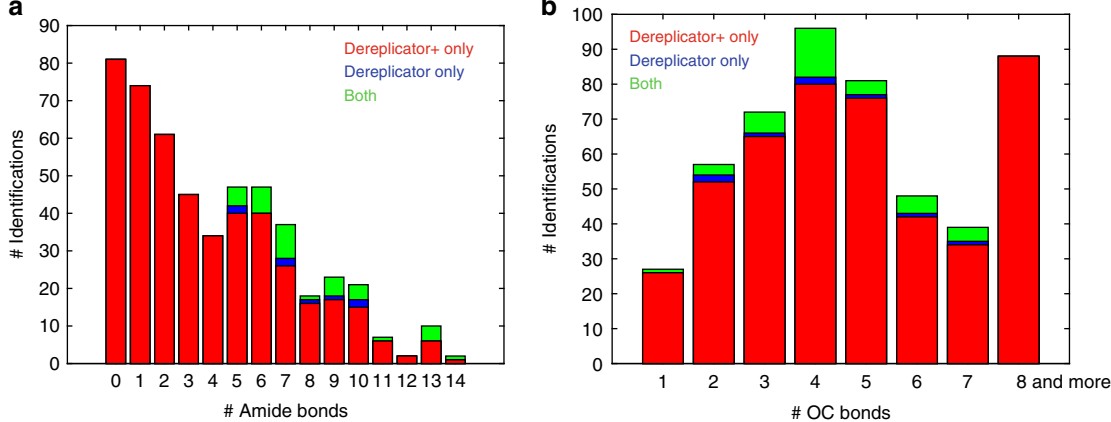

**Fig. 2** The distribution of the number chemical bonds for identifications. The distribution of the number of **a** amide bonds and **b** C–O bonds in the compounds identified at 1% FDR by DEREPLICATOR+ only (shown in red), DEREPLICATOR only (shown in blue), and both (shown in green) in Spectra$_{ActiSeq}$. Over 33% (155 out of 464) of DEREPLICATOR+ identifications have a single or no amide bonds

DEREPLICATOR+ correctly identified 34% (1878) compounds as the top rank prediction, while DEREPLICATOR-G and DEREPLICATOR identified 30% (1647) and 17% (964) compounds as top rank prediction. To break ties for matches with the same score, we took a pessimistic approach, assigning the lowest rank to the correct prediction. Since each spectrum was compared with 1235 structures on average, a random predictor would correctly identify only 5473/1235 = 4.4 spectra on average.

We reanalyzed DEREPLICATOR+, DEREPLICATOR-G, and DEREPLICATOR on Spectra$_{Library}$, reducing the precursor mass threshold from 0.5 Da to 0.02 Da. This reduced the search space from 1235 candidate structures to only 24 candidate structures per spectrum, and increased the number of correct identifications to 44% (2414), 40% (2216), and 25% (1362) for DEREPLICATOR +, DEREPLICATOR-G, and DEREPLICATOR, respectively (Fig. 3 and Supplementary Data 6). DEREPLICATOR+ identified 32% more PNPs, and overall 77% more compounds than DEREPLICATOR at 0.02 Da tolerance. MS-FINDER[28] identified 20% of compounds from Spectra$_{Library}$ as top compounds, 27% as top three, and 38% as top 10 compounds using a precursor and product ion tolerance of 0.02 Da similar to what we used for DEREPLICATOR+ (Supplementary Data 7).

We further observed a positive correlation between the success rate of DEREPLICATOR+ in identifying the correct compound as the top-scoring MSM, and the total number of oxygen–carbon (O–C) and nitrogen–carbon (N–C) bonds in the structure. For example, among compounds with 1–10 O–C/N–C bonds, 21% were correctly identified as the top-scoring MSMs, while the success rate increased to 38% for compounds with 11–20 O–C/ N–C bonds, and 55% for compounds with 21–30 O–C/N–C bonds.

Among DEREPLICATOR+ identifications of spectra from Spectra$_{Library}$ with DNP classification available (224 compounds), 28% (63) are classified as terpenes, 14% (32) are alkaloids, 13% (29) as flavonoids, 13% (28) as peptides, 8% (18) as aliphatic natural products, 8% (18) as simple aromatic natural products, 5% (11) as benzopyranoids, 4% (9) as steroids, 2% (5) as carbohydrates, 2% (5) as oxygen heterocycles, 1% (2) as lignans, 1% (2) as polycyclic aromatic natural products, 0.5% (1) as PKs, and 0.5% (1) as benzofuranoids (Supplementary Data 6).

At 3% FDR threshold, DEREPLICATOR+ identifies 3087 spectra in Spectra$_{Library}$, where for 1802 spectra the correct compound is ranked one, for 812 spectra the rank is two or three, and for 473 spectra the rank is four or above. If we define success as the prediction of the correct compound with rank 1,

DEREPLICATOR+ has a recall rate of 33% (1802 out of 5473) and precision rate of 58% (1802 out of 3087). If we define success as the prediction of the correct compound among top 3 identifications, DEREPLICATOR+ has a recall rate of 48% (2614 out of 5473) and precision rate of 85% (2614 out of 3087). DEPLICATOR+ failed to correctly identify 67% (3671 out of 5473) spectra in the Spectral$_{Library}$ dataset even though true-positive structures were present in the database.

Supplementary Note 3 and Supplementary Data 6, 8, and 9 describe the performance of DEREPLICATOR+ on isomer compounds, adducts, and negatively charges spectra.

**Identifying heterocyst glycolipid from Lichen spectra.** DERE-PLICATOR+ identified 21 metabolites in the Spectra$_{Lichen}$ dataset at 1% FDR (Supplementary Data 10). One of these metabolites, heterocyst glycolipid, was shown to be consistent with the genetic capacity detected in the lichen[39].

**Identifying almiramides from Cyanobacteria spectra.** DERE-PLICATOR+ identified 791 distinct compounds (10,375 MSMs) at 1% FDR corresponding to the score threshold of 6 (Supplementary Data 11). DEREPLICATOR identified 64 PNPs at the p value threshold of $10^{-7}$ and 1% FDR[32].

Almiramide is a linear hybrid NRP-PK with a 2-methyloct-7-enoic PK tail and amino acid sequence of Methyl-Val, Methyl-Val, Val, Ala, and Methyl-Phe isolated from *Lyngbya majuscula* PAB04NOV05-7[48]. DEREPLICATOR+ run on Spectra$_{Cyan}$ identified almiramide with 0% FDR not only in PAB04NOV05-7, but also in PAB4NOV05-1, PAP27OCT08-2, PAB9APR05-6, and PAP29JUN07-2. DEREPLICATOR failed to identify this NRP-PK at 1% FDR.

Among these datasets, metagenomics data for PAP27OCT08-2 is available. After assembling the short reads using metaSPAdes[40] and genome mining using antiSMASH[49], one of the NRPS-PKS gene clusters is assigned as the putative almiramide biosynthetic gene cluster based on the amino acid sequence and the PK genes (Supplementary Figure 5 and Supplementary Table 2). Almir-amides PK tail, 2-methyloct-7-enoic, is identical to the PK tail of jamaicamide, which was shown to be synthesized by genes *JamF*– *JamL* in the jamaicamide gene cluster[50], and these genes are very similar to genes in the almiramide gene cluster. Moreover adenylation domains in this putative gene clusters encode for two methylated phenylalaline, an alanine and a methylated alanine. Although this computational analysis points to a putative BGC

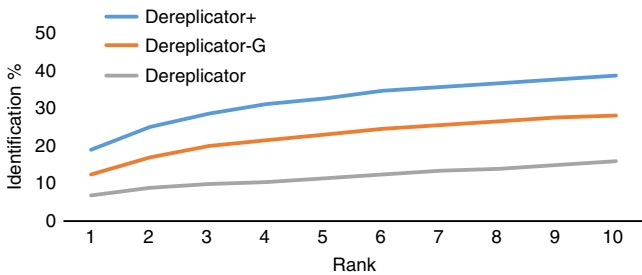

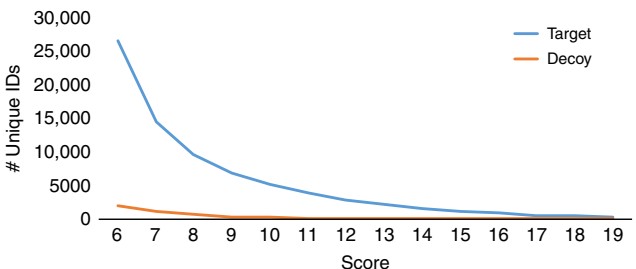

**Fig. 3** Searching 5473 MS/MS from Spectra_Library against a library composed of DNP and GNPS spectral library (86,665 compounds). Allowing for a precursor mass tolerance of 5 Da, DEREPLICATOR+ correctly identified 19% (1048) compounds as top rank identification, while DEREPLICATOR-G and DEREPLICATOR identified 12% (685) and 7% (384) compounds as top rank, respectively

**Fig. 4** The number of matches to the target and decoy databases for the Spectra_GNPS dataset. At 1% FDR (score threshold of 12), DEREPLICATOR+ identified 5336 unique compounds (410,539 MSMs) in target and 67 unique compounds (3106 MSMs) in the decoy database

for almiramide, it would benefit from a follow-up experimental validation.

**Identifying spectra from GNPS dataset.** DEREPLICATOR+ identified 410,539 spectra (0.2% of all spectra in Spectra_GNPS) representing 5336 compounds at 1% FDR and the score threshold of 12 in this dataset (Supplementary Data 12). Sixty-seven compounds (3106 spectra) were found in the decoy database at the same FDR. DEREPLICATOR identified 41,403 spectra representing 974 compounds, while DEREPLICATOR-G identified 82,260 spectra representing 1417 compounds at 1% FDR. Figure 4 shows the number of compounds identified by DEREPLICATOR+ in Spectra_GNPS at different score thresholds. Among these 5336 identifications, 643 (12%) have mass below 400, 2421 (45%) have masses between 400 to 800, 1425 (26%) have masses between 800 to 1200 Da, and 847 (15%) have masses above 1200 Da (Supplementary Figure 6).

After removing spectra with less than five peaks from Spectra_GNPS, MS-Cluster[51] partitioned it into 1,578,409 clusters. For 37,436 (2.3%) of those clusters, DEREPLICATOR+ identified at least one of the spectra in the cluster at 1% FDR (111 clusters have decoy identifications). Spectra_GNPS forms 37,750 spectral families in the spectral network[37], and DEREPLICATOR+ identified at least one member among 18,561 (49%) of these spectral families at 1% FDR (110 families have decoy identifications).

ClassyFire classified all compounds found by DEREPLICATOR as peptides. For DEREPLICATOR+ identifications, 2913 (55%) were classified as peptides, 1595 (30%) as lipids, 272 (5%) as organic oxygen compounds, 160 (3%) as organoheterocyclic compounds, 155 (3%) as PKs, and 133 (2.5%) as benzenoids. Figure 5 shows the distribution of compounds from different classes in the DEREPLICATOR identifications, DEREPLICATOR+ identifications, and the entire AntiMarin.

Among 5336 DEREPLICATOR+ identifications in Spectra_GNPS, 692 have class annotations in the DNP database. Among them, 227 (32.8%) are terpenoids, 150 (21.6%) are aliphatic natural products, 101 (14.5%) are alkaloids, 69 (9.9%) are peptides, 39 (5.6%) are simple aromatic natural products, 31 (4.4%) are steroids, 21 (3.0%) are flavonoids, 14 (2.0%) are benzopyranoids, 12 (1.7%) are lignans, 7 (1.0%) are oxygen heterocycles, 7 (1.0%) are oxygen heterocycles, 5 (0.7%) are carbohydrates, 5 (0.7%) are polypyrroles, and 3 (0.5%) are PKs.

To compare performance of DEREPLICATOR+, DEREPLICATOR-G, and DEREPLICATOR on peptides, we classified all antimarin compounds into five categories, (i) non-peptides with three or less amino acids (79%), (ii) linear peptides (7%), (iii)

cyclic peptides (3%), (iv) branch-cyclic peptides (2%), and (v) more complicated peptides (7%). Among 5336 DEREPLICATOR+ identifications, 1981 are non-peptides, 1331 are linear, 807 are cyclic, 706 are branch-cyclic, and 525 are peptides with more complicated structures. Among 1417 identifications by DEREPLICATOR-G, 175 are non-peptide, 143 are linear peptides, 607 are cyclic peptides, 369 are branch-cyclic peptides, and 123 are peptides with more complicate structures. Among 974 identifications by DEREPLICATOR, 133 are linear peptides, 487 are cyclic peptides, 298 are branch-cyclic peptides, and 56 are peptides with more complicate structures.

As an example of how DEREPLICATOR+ improves on DEREPLICATOR on peptides, we analyzed antrimycin A[52], a peptide identified by DEREPLICATOR+ at 0% FDR in actinobacteria *Kitasatospora cystargynea* RLe10 (MSV000080284), but missed by DEREPLICATOR at 3% FDR. To distinguish between the effect of addition of O–C and carbon–carbon (C–C) bonds to our model, and fragmentation graph, we introduced a scoring DEREPLICATOR-PEP-FG, where only N–C bonds up to depth three are fragmented. DEREPLICATOR+, DEREPLICATOR-G, DEREPLICATOR-PEP-FG, and DEREPLICATOR assigned scores 27, 8, 8, and 5 to the PSM formed by antrimycin A. DEREPLICATOR-PEP-FG annotated three additional internal ions as compared to DEREPLICATOR, consistent with the previous annotations of the antrimycin A spectra (Supplementary Figure 3)[53] . MS-DPR[42] assigned a $p$ value of $1 \times 10^{-6}$ and $2 \times 10^{-10}$ to the PSMs formed by antrimycin using DEREPLICATOR and DEREPLICATOR-PEP-FG scoring, respectively.

Figure 6 shows the fraction of DEREPLICATOR+ identifications at 1% FDR in Spectra_Cyan, Spectra_Acti, Spectra_Fungi, Spectra_Pseudo, and Spectra_GNPS produced by various microbial sources. Majority of compounds identified in Spectra_Cyan, Spectra_Acti, and Spectra_Fungi come from Cyanobacteria, *Actinomyces*, and Fungi sources, respectively. In the case of Spectra_Pseudo, *Pseudomonas* is the second major source after *Bacillus*, due to the presence of bacillus molecules in the growth media used for this dataset.

Among all spectra in Spectra_GNPS, 72.7% (230.1 million) are from the time-of-flight (TOF) instruments, while 18.1% (57.5 million) are from the Orbitrap instruments. Among identifications in Spectra_GNPS, 4.4 million (62.0%) are from the TOF instruments and 2.3 million (32.8%) are from the Orbitrap instruments (Supplementary Data 13).

## Discussion

The GNPS molecular networking project has enabled searches of mass spectra against spectral libraries to identify known natural products and discover their variants. However, while spectra from

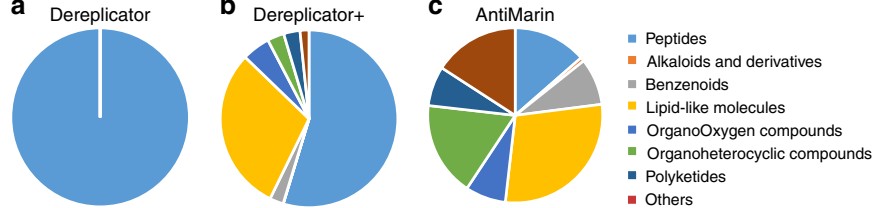

**Fig. 5** The distribution of compounds from different classes in GNPS. **a** The distribution of compounds from different classes among 739 identifications from DEREPLICATOR. **b** The distribution of compounds from different classes among 5336 identifications from DEREPLICATOR+. **c** The distribution of compounds from different classes among all 60,908 AntiMarin compounds. All compounds were classified with ClassyFire[43]

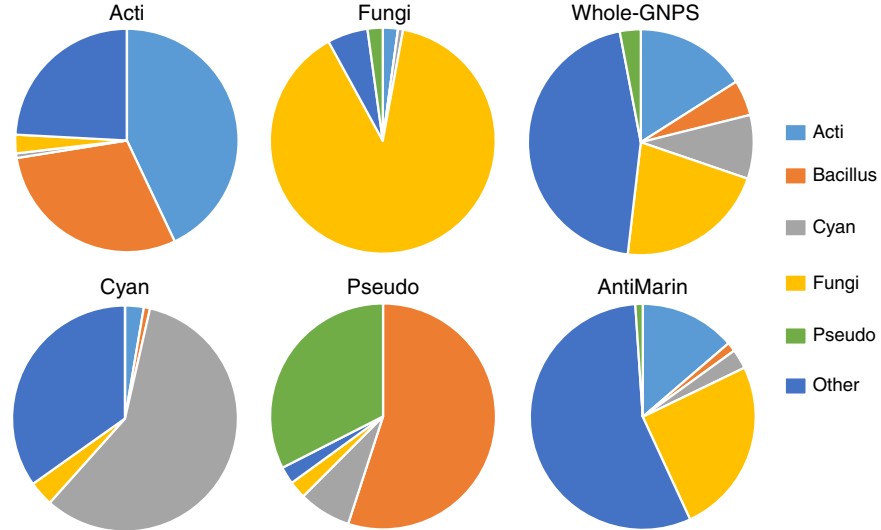

**Fig. 6** The fractions of DEREPLICATOR+ identifications from various sources in Spectra$_{Cyan}$ (112 compounds), Spectra$_{Acti}$ (149 compounds), Spectra$_{Fungi}$ (138 compounds), and Spectra$_{Pseudo}$ (40 compounds). Majority of compounds identified in Spectra$_{Cyan}$, Spectra$_{Acti}$, and Spectra$_{Fungi}$ come from Cyanobacteria, *Actinomyces*, and Fungi sources, respectively. In the case of Spectra$_{Pseudo}$, *Pseudomonas* is the second major source after *Bacillus*, due to contamination. The fractions of compounds from different sources in Spectra$_{GNPS}$ (5336 compounds) and AntiMarin (60,908 compounds) are also shown

GNPS represent a gold mine for future natural products discovery, their interpretation remains challenging. Currently, the GNPS spectral library, the most comprehensive spectral library of natural products, has <10,000 compounds. The vast majority of GNPS spectra have evaded all attempts to interpret them, indicating that there exists a large dark matter of metabolomics[4].

Currently, only 0.2% of spectra from GNPS are identified by spectral library search. Many of the still unidentified spectra are likely formed by known molecules present in chemical structure databases such as PubChem. Identifying the spectra of the compounds which are present in chemical structure libraries but absent in spectral libraries requires the development of algorithms for matching spectra of natural products against chemical databases.

We recently developed DEREPLICATOR and VarQuest[32,54] tools for identifying PNPs and their variants through database search of mass spectra. However, these tools are not designed for identification of other types of natural products such as PKs and lipids. Generalizing them to other types of natural products requires both generalizing from amide bonds to other types of bonds that usually break during tandem mass spectrometry (e.g., C–O bonds and C–C bonds) and also handling the sequential process of a compound fragmentation as a fragmentation graph. Our results show that generalized bond breakage and the sequential fragmentation are both crucial in enabling DEREPLICATOR+ to identify 77% more compounds (consisting of non-peptide metabolites and mixed peptide-PKs) that were missed by

DEREPLICATOR. Moreover, ≈33% of compounds identified by DEREPLICATOR+ have either a single or no amide bond.

In contrast to existing database search tools in metabolomics[23–36], DEREPLICATOR+ is the first database search tool for natural products that can (i) search the entire GNPS molecular networking infrastructure against large databases of chemical structures, and (ii) identify variants of known metabolites using molecular networking[37,44]. Similar to all the existing metabolomics tools, DEREPLICATOR+ is not applicable to all the classes of compounds. DEREPLICATOR+ greatly improves on the existing tools for PNPs and enables identifying the important class of peptides-PKs and PKs.

Currently DEREPLICATOR+ uses a simple shared peaks scoring to match spectra against fragmentation graphs, and computing FDR for the found identifications. This simple scoring scheme results in a high false-negative rate. Meanwhile, DEREPLICATOR+ enables automatic collection of large metabolite spectra libraries by searching billions of spectra, similar to those in proteomics. This pave the path for machine learning methods to improve scoring scheme and enhance the false-negative rates.

## Methods
**Overview of the DEREPLICATOR algorithm**. DEREPLICATOR[32] generates the theoretical spectrum of a PNP by first constructing a peptide graph, where each node is an amino acid and each edge is a peptide bond. Afterwards, it considers various fragmentations of the peptide graph by removing each possible *bridge* or *2-cut* (each such removal breaks the peptide graph into two connected components). DEREPLICATOR generates the theoretical spectrum of a peptide as the set of

masses of all resulting connected components, where the mass of a component is defined as the total mass of its constituent amino acids. Afterwards, it compares each experimental spectrum against the theoretical spectra of all PNPs in a database and finds the best scoring PSM. The statistically significant PSMs (as defined by their $p$ values) reveal identified PNPs.

**Outline of DEREPLICATOR+ algorithm**. While DEREPLICATOR was shown to accurately identify spectra of PNPs, this approach is not applicable to other types of metabolites that go through multiple fragmentations at different bonds in a mass spectrometer. To address this complication, DEREPLICATOR+ introduces the concept of the fragmentation graph. Below, we describe various steps of the DEREPLICATOR+ algorithm for identifying spectra of all metabolites.

**Constructing metabolite graphs**. Supplementary Tables 3 and 4 show prevalence of different atoms and bonds in natural products from the AntiMarin database. In addition to N–C bonds in peptides, metabolites often break at O–C and C–C bonds. We thus refer to N–C, O–C, and C–C bonds as metabolite bonds. Disconnecting all metabolite bonds in a metabolite breaks it into connected component that form the nodes of the metabolite graph labeled by their elemental composition (the mass of the node is defined as the total mass of elements in the connected component). Edges in the metabolite graph correspond to metabolite bonds (Fig. 7). The theoretical spectrum of a metabolite graph is defined by the masses of all connected subgraphs of the metabolite graph formed by disconnecting bridges and 2-cuts, and the DEREPLICATOR-G score refers to the count of shared peaks between the experimental spectrum and the theoretical spectrum of the metabolite graph within a tolerance threshold. We further classify all 2-cuts into feasible (formed by either C–N or C–O bonds) and infeasible (containing at least one C–C bond).

**Generating fragmentation graphs**. Since metabolites often go through multiple fragmentations at different bonds during tandem mass spectrometry, theoretical spectra formed by all bridges and 2-cuts of the metabolite graphs poorly model real spectra. We thus model a theoretical spectrum of a metabolite as a fragmentation graph originally introduced for de novo sequencing of PNPs by multistage mass spectrometry[55]. Note that our concept of the fragmentation graph (constructed based on the chemical structure of a known metabolite) differs from the fragmentation trees introduced in Rasche et al.[56], which are constructed based solely on mass spectra to characterize chemical formula and potential substructure motifs of the metabolites[29]. Constructing fragmentation graphs solely from structures has computational advantages over the relatively slow fragmentation tree approach since it allows one to precompute fragmentation graphs for all structures in a single scan of a chemical database, thus enabling fast analysis of large spectral datasets. The additional advantage of constructing fragmentation graphs from chemical structures is that, in contrast to fragmentation trees[56], this method can capture substructures absent from the public substructure databases such as PubChem CACTVS and Klekota–Roth[57].

Disconnecting a bridge or a 2-cut in a graph $G$ breaks it into connected components that we call descendants of $G$. Given a metabolite graph $G$, its fragmentation graph is defined as follows. The nodes of the graph are defined as all possible connected subgraphs of $G$ and the edges of the graph are defined as directed edges connecting subgraphs to their descendants. We define the complete metabolite as the root of the graph, and define the depth of each node in the graph as the length of the shortest path from the root to this node (Fig. 8). Since mass spectrometry experiments with conventional collision energies usually do not produce many fragment ions corresponding to 3-cuts, we do not analyze 3-connected metabolite graphs. Only 70 metabolite graphs arising from compounds in AntiMarin (0.1%) are 3-connected. We further trim the fragmentation graph to eliminate nodes that correspond to unrealistic fragmentation rules.

Given a parameter $k$, we use the following constraints to filter down the nodes in the fragmentation graphs: (i) all nodes with depths exceeding $k$ are removed from the graph, (ii) only nodes with a path from the root consisting of maximum one 2-cut fragmentation are retained in the graph, (iii) 2-cut fragmentations are limited to feasible 2-cuts, and (iv) only nodes with a path from the root consisting of maximum one C–C bond fragmentation are retained in the graph. We use constraints (iii) and (iv) because, while the C–C bonds are common in natural products, they are less likely to break as compared to the C–N and C–O bonds. We evaluated the rationality of these rules by comparing 14 different fragmentation models on the Spectra_{library} dataset based on the total log likelihood of the known MSMs in comparison to the null fragmentation model that assumes that all the MSMs are random (Supplementary Note 4 and Supplementary Data 14). Our results show that, the optimal value of $k$ is 6 in condition (i), and that conditions (ii)–(iv) represent rational rules to prune the fragmentation graph.

We constructed the fragmentation graph of heterocyst glycolipid[58] with chemical formula $C_{32}O_8H_{64}$. When considering only constraint (i), this fragmentation graph has 99 nodes at depth 1, 2721 nodes at depth 2, and 7216 nodes at depth 3. After trimming the graph with constraints (ii), (iii), and (iv), it has 69 nodes at depth 1, 632 nodes at depth 2, and 1182 nodes at depth 3. Note that there might be multiple nodes with identical masses and chemical formulas that are produced by different bridges and 2-cuts in the fragmentation graph.

**Generating decoy fragmentation graphs**. Supplementary Figure 7 shows the histogram of masses of all experimental peaks in the Spectra_{Lichen} dataset[39], and histogram of masses of all peaks from the theoretical spectra of metabolites in the AntiMarin database. We normalize the histograms of masses of experimental peaks to turn it into probability distributions and refer to the probability of an experimental peak being in the interval $[x, y]$ as $F(x, y)$. Similarly to generating decoy spectra for spectral library searches[59], DEREPLICATOR+ generates a decoy fragmentation graph for each target fragmentation graph. This is achieved by fixing the decoy structure identical to the target, and proceeding in a breadth-first manner by assigning a mass to each node in the decoy fragmentation graph as follows. The mass of the root is equal to the total mass of the metabolite, and for each node $v$, DEREPLICATOR+ samples mass($v$) from the range [0,Mass(Parent($v$))] of the distribution of all theoretical peaks, where Parent($v$) is parent of the node $v$. In cases where a node has multiple parents in a graph, we consider the parent with minimum Mass (Supplementary Figure 8). Because we use fragmentation graphs from all AntiMarin compounds to learn the distribution from which we sample decoy fragmentation graphs, it captures the masses common in natural products.

**Annotating fragmentation graphs by spectra and scoring MSMs**. Similar to MS-Cluster and Molecular Networking approaches[37,44], DEREPLICATOR+ processes all spectra by retaining only six top intensity peaks in each window of size 50 Da. Given a spectrum, we annotate the nodes in a fragmentation graph in the breadth-first manner, starting from nodes at depth 1 (the root is assumed to be annotated). We say that a node is annotated by a spectrum, if (i) at least one of its direct ancestors is annotated, and (ii) its mass is explained by a peak in the spectrum. Given a metabolite $M$ and spectrum $S$, we define score($M,S$) as the number of unique masses in $S$ that annotate a node in the fragmentation graph of $M$. We also define score$_i$($M,S$) as the number of unique masses in $S$ that annotate a node at the depth $i$ and lower in the fragmentation graph of $M$. Using a threshold of 0.02 Da on ion detections for the spectrum of heterocyst glycolipid with 44 peaks, 2 peaks are annotated at depth 1, 3 peaks at depth 2, and 9 peaks at depth 3 (Fig. 9).

**Computing the statistical significance of MSMs and evaluating FDR**. We refer to the set of all depth 1 nodes of the fragmentation graph of a metabolite $M$ as Children(root) and the number of peaks in $S$ as $|S|$. Given an MSM formed by a metabolite $M$ and a spectrum $S$, we estimate its statistical significance with respect to score$_1$($M,S$) based on the following probabilistic model for a random MSM.

All the nodes from Children(root) are annotated independently with the probabilities as defined below. The score is calculated as the number of the annotated peaks and the probability of the random event of peak annotation is computed from the empirical distribution $F$ of all experimental peaks shown in Supplementary Figure 7 in the range from mass 0 to the precursor mass of the spectrum $S$. The statistical significance of a match between metabolite $M$ and spectrum $S$ is defined as the ratio of random MSM scores exceeding or equal to score$_1$($M,S$).

The probability $p$ that a random experimental peak sampled from range [0, Mass($M$)] of the experimental distribution annotates a theoretical peak $m$ within tolerance $\delta$ is computed as (Supplementary Figure 9):

$$p(m) = F(m - \delta, m + \delta). \tag{1}$$

Given $|S|$ experimental peaks, the probability that at least one of them annotates a theoretical peak $m$ can be computed as:

$$q(m) = 1 - (1 - p(m))^{|S|} \tag{2}$$

and the probability of having exactly $s$ matches between metabolite graph $M$ and spectrum $S$ has a Poisson binomial distribution (note that due to our probability model definition the annotation events are independent):

$$P(\text{number of match} = s)$$
$$= \sum_{\text{all subsets } I \text{ of peaks in Children(root) of size } s} \prod_{m \in I} q(m) \prod_{m \notin I} (1 - q(m)). \tag{3}$$

Since there are |Children(root)| terms in the Poisson binomial distribution, it is not feasible to compute it in a brute force manner for large theoretical spectra and scores. However, the Poisson binomial distribution at value $s$ is equal to the coefficient of $Z^s$ in the following generating polynomial[60]:

$$\prod_{m \in \text{Children(root)}} (1 - q(m) + q(m)Z) \tag{4}$$

and this formula leads to an efficient approach for computing the $p$ value of score$_1$.

This procedure can be generalized to scores at higher depths (score$_i$ for $i > 1$) in the case of fragmentation graphs with tree structures. For leaf node $u$ with mass $m_u$,

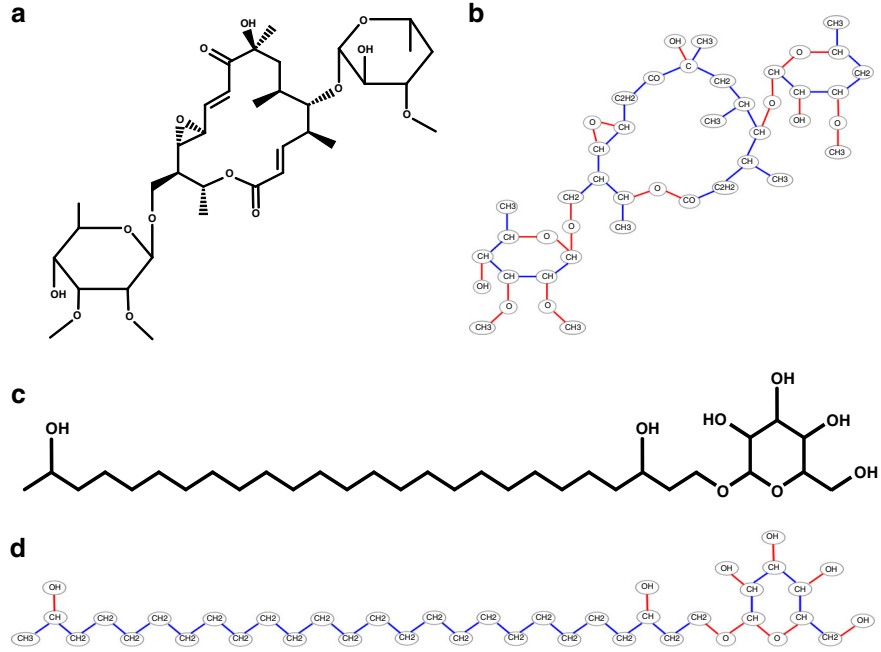

**Fig. 7** From metabolite structure to metabolite graph. **a** Structure of a polyketide chalcomycin, **b** the metabolite graph of chalcomycin with 43 nodes and 46 edges, **c** structure of a terpene heterocyst glycolipid, and **d** metabolite graph of heterocyst glycolipid with 40 nodes and 40 edges. Each node in the metabolite graph is a connected component of the structure after dissociating all metabolite bonds. C–C bonds are shown in blue and C–O bonds are shown in red. The metabolite graph of chalcomycin has 43 nodes and 46 edges, including 18 bridges, 8 feasible 2-cuts, and 20 infeasible 2-cuts. The metabolite graph of heterocyst has 40 nodes and 40 edges including 34 bridges, 2 feasible 2-cuts, and 4 infeasible 2-cuts

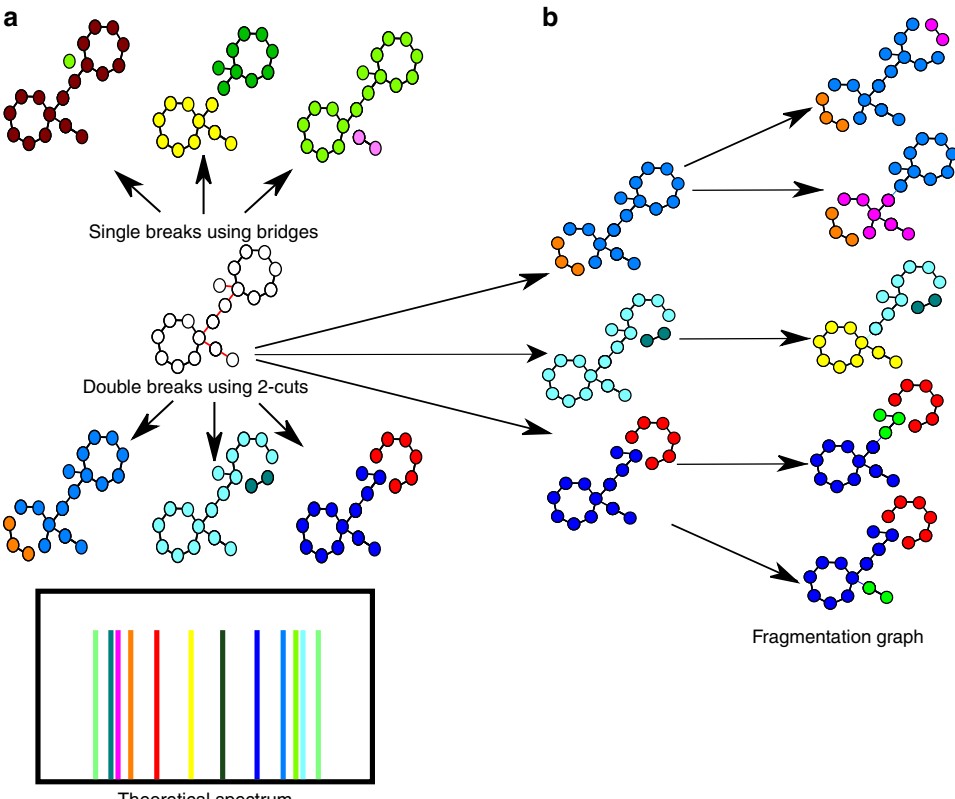

**Fig. 8** Generating the fragmentation graph. **a** DEREPLICATOR[32] generates the *theoretical spectrum* of a PNP by first constructing a *peptide graph*, where each node is an amino acid and each edge is a peptide bond. Afterwards, it considers various fragmentations of the peptide graph by removing each possible bridge and 2-cut (each such removal breaks the peptide graph into two connected components). DEREPLICATOR generates the theoretical spectrum of a peptide as the set of masses of all resulting connected components, where the mass of a component is defined as the total mass of its constituent amino acids. **b** DEREPLICATOR+ sequentially breaks the metabolite graph at *multiple* bridges and 2-cuts, and models a theoretical spectrum of a metabolite as a fragmentation graph. Only a small subgraph of the fragmentation graph is shown

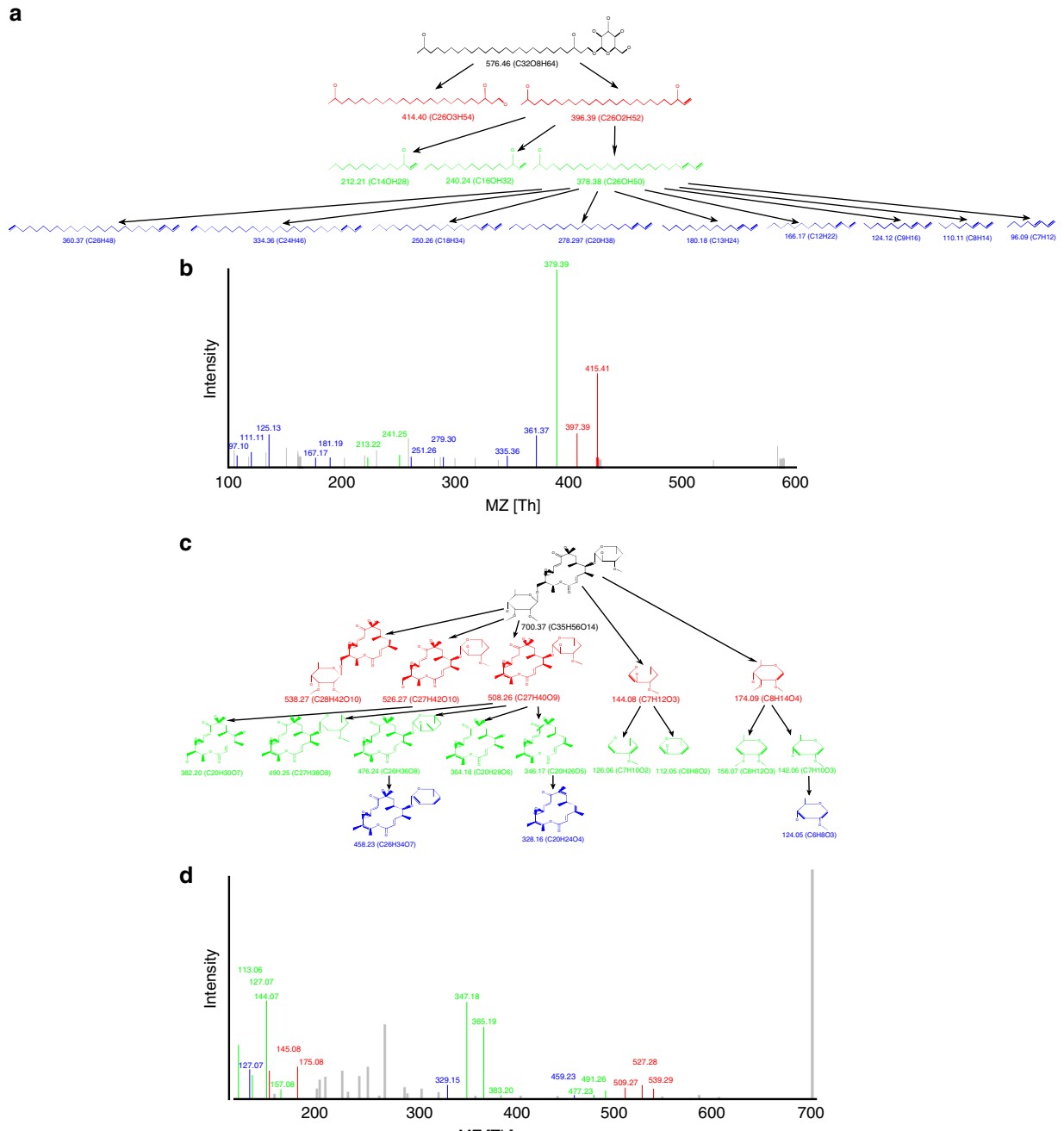

**Fig. 9** Annotating the fragmentation graph. The MSM formed by the fragmentation graph of **a** heterocyst glycolipid, and **b** its spectrum. While only 2 and 3 peaks out of 44 peaks in heterocyst glycolipid mass spectra get annotated at depths 1 and 2, respectively (shown in red and green), 9 peaks get annotated at depth 3 (shown in blue), resulting in a score of $2 + 3 + 9 = 14$. **c** The MSM formed by the fragmentation graph of **c** chalcomycin, and **d** its mass spectrum. 5, 9, and 3 out of 42 peaks in chalcomycin spectrum are annotated at depth 1, 2, and 3, respectively, resulting in a score of $5 + 9 + 3 = 17$

we define the generating polynomial $H_u$ as

$$H_u(Z) = 1 - q(m_u) + q(m_u) \cdot Z \tag{5}$$

and for non-leaf nodes, we define:

$$H_u(Z) = 1 - q(m_u) + q(m_u) \cdot Z \cdot \prod_{v \in \text{Children}(u)} H_v(Z). \tag{6}$$

Following this recursion, $H_{\text{root}}$ contains probability distribution of the score. In general case of non-tree fragmentation graphs, the proposed recursion provides a rough approximation in the considered probabilistic model due to statistical dependence of annotation events in vertices sharing descenders. For unbiased probability estimation Markov Chain Monte Carlo methods can be utilized.

In the case of heterocyst glycolipid discovered in Spectra$_{\text{Lichen}}$, the probability that 2 or more out of 44 peaks in its spectrum are explained by children of the root with a tolerance of 0.02 Da is 0.026. For a child of the root with mass 396.39 Da, the probability that 3 or more of its children are annotated is 0.0003 Da. For a child of this node with mass 378.38, the probability that 9 or more of its children are annotated is $3 \times 10^{-15}$.

**Enlarging the set of MSMs using molecular networking**. The concept of spectral networks[37] (also known as molecular networks[44] in the field of natural products) was introduced to reveal spectra of related peptides within a proteomic dataset without knowing what these peptides are. While these networks were first introduced for linear peptides, they were later generalized to cyclic peptides and metabolites[44,61]. Nodes in a molecular network correspond to spectra, while edges

connect spectra that are generated from related metabolites (e.g., metabolites differing by a single variation). The variations that are captured by molecular networks help to infer mutations, modifications (such as oxidation and acetylation), or adducts (such as sodium and potassium).

DEREPLICATOR+ constructs the molecular network of all spectra and selects connected components with at least one identified spectrum. Using the MSM corresponding to this spectrum, it annotates variants of identified metabolites. Supplementary Figure 10 shows the molecular network of the chalcolmycin family identified by DEREPLICATOR+.

**Versions of DEREPLICATOR+.** To analyze the differences between DEREPLICATOR and DEREPLICATOR+, we analyzed the Spectra$_{ActiSeq}$ dataset using three versions of DEREPLICATOR+ described below. DEREPLICATOR-CN is a version of DEREPLICATOR, where all C–N bonds are cut rather than amide bonds only. DEREPLICATOR-CN-CO is a version of DEREPLICATOR, where all C–N and all C–O bonds are cut. In DEREPLICATOR-G all C–N, C–O, and C–C bonds are cut. DEREPLICATOR+ differs with DEREPLICATOR-G in consideration of depth 2 and depth 3 fragmentations in addition to depth 1.

**Code availability**. DEREPLICATOR+ is available as both a stand-alone tool (http://mohimanilab.cbd.cmu.edu/software/) and a web application (http://gnps.ucsd.edu/ProteoSAFe/static/gnps-theoretical.jsp).

## Data availability

All datasets analyzed in this study are available through GNPS infrastructures with access codes available in the Supplementary Table 1. All other data supporting the findings of this study are available from the corresponding author upon reasonable request.

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

## Acknowledgements

We would like to thank Dr. Hiroshi Tsugawa for providing guidelines on running MS-FINDER against custom databases. We would also like to thank Dr. Tobias Find and Dr. Oliver Fiehn for providing access to the chemical structure library of LipidBlast. The work of H.M. was supported by the start up package from Carnegie Mellon University. The work of P.C.D., and P.A.P. was supported by the US National Institutes of Health grant 2-P41-GM103484. L.-F.N. and P.C.D. are supported by GM097509. A.G., A.M., A. S., A.K., and P.A.P. were supported by Russian Science Foundation (grant 14-50-00069).

## Author contributions

H.M., A.G., A.S., A.M., and A.K. implemented DEREPLICATOR algorithm. H.M., E.S. and L.C. performed the analysis. L.-F.N. analyzed the results of the analysis. H.M., P.C.D. and P.A.P. designed and directed the work. H.M. and P.A.P. wrote the manuscript.

## Additional information

**Competing interests:** P.A.P. has an equity interest in Digital Proteomics, LLC, a company that may potentially benefit from the research results. The terms of this arrangement have been reviewed and approved by the University of California, San Diego in accordance with its conflict of interest policies. The remaining authors declare no competing interests.

