## [Peer Review File · Nature Communications]

Reviewers' comments:

Reviewer #1 (Remarks to the Author):

DEREPLICATOR+: In silico Identification of Natural Products Through Database Search of Mass Spectra.

According to the authors, this manuscript focuses on the improvement of a dereplication tool (Dereplicator+), originally developed for identification of peptides of natural product (Dereplicator) and extended here for different classes of secondary metabolites.

In an overview, the strategy uses the same sequential procedure of the previous article published in Nature Chemical Biology (Nature Chemical Biology volume 13, pages 30-37, 2017). However, it brings an innovative approach that is the use of chemical bond cleavage (based on the normal distribution of databases) for the generation of decoy structures (fragmentation patterns) and therefore use them as a scoring criterion for the matching among metabolites and spectra. Molecular Networking was also used to amplify the identification of correlated metabolites that are associated with the annotated spectrum (filtered by MSMs and by "statistical significance" - p-values and FDR).

Although it is a potentially promising tool for dereplication, some concerns should be addressed in relation to the thresholds used. In general, the five thresholds (in the section: Generating Fragmentation Graphs) should be better in terms of benefits and limitations. Is there some literature that can explain those five heuristic rules? For instance, the use of up to "2-cuts" on which was based on the AntiMarin database. What is the correspondence of the pattern of fragmentation between substances of marine origin and those obtained from terrestrial origin, that is, what is the correspondence between the classes of marine and terrestrial biosynthesis? Also, in cases where multiglycosylated or multitotoxylated compounds such as glycosylated flavonoids are present, the depth number can easily exceed 3 levels.

Thus, I recommend the publication of the document, although the authors can provide a better explanation of the limitations found in the process of identification of secondary metabolites in the manuscript. I believe that the documentation on the in silico tools in the GNPS already presents some of the limitations of this dereplication tool - step 8 (<https://bix-lab.ucsd.edu/display/Public/Insilico+Peptidic+Natural+Products+Dereplicator+Documentation>).

Some specific suggestions and mistyping encountered in the manuscript.

In "Development of chemical structure databases such as PubChem (...)", the authors can include others appropriated compounds databases such as ChemBank, ChEMBL, ChEBI, DrugBank, NuBBEDB, Chempider etc. (page: 1).

In the text are encountered two forms of AntiMarin and Antimarin database, please choose one of them;

In "Analyzing Spectralibrary dataset" the word precursor is incorreceted; Also in "Analyzing Spectralibrary dataset we suggest to replace reran by reanalyzed (page: 5)

In "Generating decoy fragmentation graphs" in methods, remove one of the "into" present in first line on page 9.

In "Annotating fragmentation graphs by spectra and scoring MSMs" also in methods sections correct the word spectrum (page 9).

In addition, I suggest including an example for evaluation of the dereplicator+ tool for peptide and a secondary metabolite on the GNPS platform.

Reviewer #2 (Remarks to the Author):

The idea of natural products dereplication focuses on identifying already known compounds in a sample so further purification and bioassay experiments can focus on the uncharacterized compounds. The paper reports a minor improvement in an existing cheminformatics approach, DEREPLICATOR which aims to computationally confirm the presence of known chemical structures in a biological sample by utilizing only mass spectral data. The parent approach is limited to peptides and authors have attempted to extend it to other chemical classes of natural products. While the paper tried to tackle a key challenge to identify non-peptide compounds in the mass spectral datasets, the proposed tool (DEREPLICATOR+) still need extensive testing and validation in order to prove its versatility, accuracy and comprehensiveness to identify those compounds. Major concerns are :

- Paper reports identification of very few chemicals from a small number of non-peptide chemical classes. This does not prove that the tool can reliably identify all the possible known chemicals of a non-peptide chemical class of natural products in a microbial sample's mass spectral dataset.
- Identification of non-peptide compounds in mass spectral datasets is routinely performed using mass spectral library searches. Authors have not compared their tool's performance against these existing approaches and prove that the tool works better than that. It needs to be demonstrated that how many compounds the tool can identify which were missed by mass spectral library searches and vice versa.
- Tool's accuracy is not demonstrated to predict substructures for fragments in available known mass spectra for non-peptide compounds in public mass spectral libraries.
- Only tool's specificity is shown but its sensitivity is also important for compound dereplication efforts. What is the confidence level that a compound suggested being absent by the tool is really not in the sample? A higher false negative rate can misguide investigators by not identifying compounds which were present in the sample and in the structural database. As the goal of the dereplication is to guide them on novel compounds, a higher false negative rate can direct them on something that is already present in structural databases. Data for both precision and recall need to be shown for the identification of non-peptide chemical classes.
- Isomeric compounds are present in mass spectral datasets for a biological specimen. Paper did not explain how the tool discriminates those compounds.
- During the electrospray ionization, a compound can generate multiple adducts and in-source fragments which can have different MS/MS spectra (<https://pubs.acs.org/doi/10.1021/acs.analchem.7b02380> and <https://www.ncbi.nlm.nih.gov/pmc/articles/PMC5750104/>) . How the tool handles these mass spectra is not explained. Theoretical spectrum of these adducts and fragments were not generated by the tool. Were they identified as different compounds ?
- Up to 95% natural products in the DNP database are below 1000Da. However the tool seems to be working better for compounds above this threshold. Tool's performance for compounds below 1000Da needs to be shown thoroughly.

Below are specific comments.

Title :

- "natural products" should be changed to "microbial natural products" as the paper does not have data for plant-specific natural products.
- Replace "in-silico identification" with "dereplication". The word "Identification" gives the impression that it can identify completely unknown compounds in a sample which is not correct. The tool starts with a compound structure database that will be searched against a mass spectral dataset to confirm only the presence of those compounds.

Abstract :

“other classes” should be replaced with exact chemical classes that the tool can identify. The main compelling idea of the paper is that the tool can go beyond peptides and lipids, I think it is important to show those classes in abstract.

Introduction :

Misleading statements :

- “most of the existing tools for metabolite identifications work best for identification of small molecule (below 500 Da) and become prohibitively time-consuming for searching large spectral datasets”. LipidBlast can identify lipids up to 2000 Da (<https://www.nature.com/articles/nmeth.2551>). And, NIST MS Search software (<http://chemdata.nist.gov/dokuwiki/doku.php?id=peptidew:mspepsearch>) can search against large spectral datasets.
- “approaches that construct in silico mass spectra by fragmenting natural peptides along specific bonds^{39,40}”. LipidBlast (ref number 40) generates in-silico spectra for lipids not peptides.

Results

Experimental datasets (MSV000078604 , MSV000078839, MSV000078568, MSV000078584)

- Provide ionization mode, liquid chromatography parameters and mass spectrometry instrument details for the experimental datasets.
- Ionization modes gives different mass spectra for the same compound. How the tool perform for both ESI (-) and ESI (+) modes ?
- The quality of mass spectra depends on the utilized instrument. How the tool performs for data acquired on different instruments ?
- HCD or CID spectra are different for several compounds. Was the tool tested that it works for both type of spectra with same accuracy?
- How it was ensured that MS/MS spectra in these experimental datasets were clean and did not contain any isobaric species or contaminant fragments (<https://www.ncbi.nlm.nih.gov/pubmed/18790129>) ? Were any quality checks and filtering were applied on the spectra before matching them against the known structures?

Structural databases (DNP and AntiMarin):

- DNP database has almost 150,000 compounds. Why only 83889 were used ? Was any filtering on the database was applied ? Tool need to recalculate the results for entire DNP database.
- DNP has 99 compounds listed for Actinomyces species. Were these compound present in the DNP-subset used in the paper.
- How tool perform for entire PubChem database? Can it identify more compounds in the experimental mass spectra datasets using PubChem? DNP is a small subset database of possible natural products. Searching against PubChem shall identify many more compounds in these mass spectral datasets.
- “Spectralibrary “ has associated chemical structures. The tool needs to generate spectra for these structures and show that how many were accurate.

Analyzing SpectraActiSeq dataset

- “66 unique compounds (148 MSMs)” indicate that 148 spectra belonging to these 66 unique compounds. Count of identified compound increased by two-fold however the count of identified spectra increased by 18 fold, why ?
- It should be stated that only 1.5% (2666/178635) of all spectra were identified.
- DNP provides the class information for compounds. ClassyFire poorly covers many natural product classes that DNP has. What were the classes for the identified compounds in the DNP database ?
- Statement “Table 1 describes 25=29-4 compounds (covering 20 PNPs, 2 polyketides, 2 terpenes, and 1 benzenoid) identified by DEREPLICATOR+.” indicates that the tool still needs improvement

to comprehensively cover non-peptide chemical classes. Just identifying one compound from a class does not prove that it should be able to identify other chemicals in that class.

- Authors have compared DEREPLICATOR+ against its parent tool. But it must be demonstrated that the tool works better than existing mass spectral matching approaches to identify non-peptide compounds in experimental mass spectral datasets. Specifically, the overlap among compounds identified using NIST17, LipidBlast and MONA databases and DEREPLICATOR+ need to be computed.
- The tool also need to be compared with the MS-FINDER software (<https://pubs.acs.org/doi/abs/10.1021/acs.analchem.6b00770>) which can also predict substructures for product ions.
- DNP lists 99 compounds for Actinomyces species. How many of them were missed by the tool ?

Analyzing Spectralibrary dataset.

- "This reduced the search space from 135 candidate structures to only 24 candidate structures per spectrum, and increased the number of identifications on the top to 44%(2414), 40%(2216) and 25%(1362) for DEREPLICATOR+, DEREPLICATOR-G and DEREPLICATOR, respectively." What classes were covered among these 2414 identified compounds ? How many of them were non-peptides ? Why 56% spectra could not be identified by the tool ?

Analyzing Spectralichen dataset and Analyzing SpectraCyan dataset.

- Which DNP classes were covered ? ClassyFire is not a relevant classification ontology for natural products.

Assigning almiramide biosynthetic gene cluster.

- It needs to be shown that the parent DEREPLICATOR tool failed to identify this compound and only DEREPLICATOR+ was able to do so. This compound is still a peptide. Did author identify any gene cluster for non-peptide natural products ?

Analyzing SpectraGNPS dataset.

- "For DEREPLICATOR+ identifications, 54.6% were classified as peptides, 29.9% as lipids, 5.1% as organic oxygen compounds, 3.0% as organoheterocyclic compounds, 2.9% as polyketides and 2.5% as benzenoids." . Which DNP classes were covered in this ? Lipids structures can be dereplicated using the LipidBlast library so the lipid identification need to be compared with results from LipidBlast annotation.
- Results on peptides can be shortened as the main point of the paper is to identify non-peptide chemical classes.

Discussion

Statement "While many of the existing approaches focus on small metabolites with mass below 1000Da³⁶, DEREPLICATOR+ works for both small and large metabolites." -is overstated and can be toned down.

- LipidBlast can identify lipids up to 2000Da.
- Molecular weights distribution of the identified compounds by dereplicator+ is not provided in the paper to support this claim. DNP database says that only 5% of its compounds are above 1000Da. It seems that a majority of natural products are below 1000 Da but the majority of the identified compounds by the tool are above 1000 Da which is a small fraction of the natural products.

Reviewer #1 (Remarks to the Author):

C1.1. DEREPLICATOR+: In silico Identification of Natural Products Through Database Search of Mass Spectra.

According to the authors, this manuscript focuses on the improvement of a dereplication tool (Dereplicator+), originally developed for identification of peptides of natural product (Dereplicator) and extended here for different classes of secondary metabolites. In an overview, the strategy uses the same sequential procedure of the previous article published in Nature Chemical Biology (Nature Chemical Biology volume 13, pages 30-37, 2017). However, it brings an innovative approach that is the use of chemical bond cleavage (based on the normal distribution of databases) for the generation of decoy structures (fragmentation patterns) and therefore use them as a scoring criterion for the matching among metabolites and spectra. Molecular Networking was also used to amplify the identification of correlated metabolites that are associated with the annotated spectrum (filtered by MSMs and by "statistical significance" - p-values and FDR).

R1.1. Thank you for kind words about our paper – we have addressed your comments below. The changes are indicated in red in the revised manuscript and the Supplement.

C1.2. Although it is a potentially promising tool for dereplication, some concerns should be addressed in relation to the thresholds used. In general, the five thresholds (in the section: Generating Fragmentation Graphs) should be better in terms of benefits and limitations. Is there some literature that can explain those five heuristic rules? For instance, the use of up to "2-cuts" on which was based on the AntiMarin database.

R1.2 We acknowledge that our approach represents a heuristic rather than a rigorously justified approach with respect to parameter choices. The only excuse we have is that in traditional proteomics, even after two decades of computational developments, some of the most popular tools (e.g., Mascot or X!Tandem) represent heuristics with unclear parameter choices.

To address the problem of selecting five thresholds, we tried 14 different fragmentation models, and compared their performance to DEREPLICATOR+. To evaluate the performances, we scored a set of known metabolite-spectrum-matches using each of the 14 models. We added the following to the main text :

We evaluated the rationality of these rules by comparing 14 different fragmentation models on the *Spectra_{library}* dataset based on the total log-likelihood of the known Metabolite-Spectrum-Matches (MSMs) in comparison to the null fragmentation model that assumes that all the MSMs are random (Supplementary Text 4 and Supplementary Table 18). Our results show the optimal value of k is 6 in condition (i), and that conditions (ii)-(iv) represent rational rules to prune the fragmentation graph.

We added the following as Supplementary Text 4 :

Supplementary Text 4. Fragmentation model selection. To evaluate the rationality of the rules and thresholds in the fragmentation model, we compared 14 fragmentation models with various parameters used in conditions (i)-(iv). The comparison is based on the log-likelihood of these models in explaining the experimental spectrum (in comparison to the null model) over a set of 5473 Metabolite-Spectrum Matches (MSMs) from the *Spectra_{library}* dataset. The models differ from each other in the chemical bonds they fragment, the maximum number of bridges and 2-cuts they allow, and whether or not they allow for the breakage of C-C 2-cuts are multiple C-C bridges. $\#Max_L$ stands for the maximum number of bridge fragmentations allowed. $\#Max_C$ stands for the maximum number of the 2-cut fragmentations allowed. $\#Max_{2L+C}$ is an upper-bound on twice the number of 2-cuts plus the number of bridges (i.e. total number of cuts in the fragmentation). $theo_{ef}$ stands for the size of the effective theoretical spectrum of a MSM, defined as peaks in the theoretical spectrum with annotated parents (the root is always annotated). $\#explained$ stands for the number of explained peaks in the experimental spectrum. For a specific model θ , p_{θ} stand for the probability of a peak in the effective theoretical spectrum being explained by the model. The null hypothesis assumes peaks are explained independent of the theoretical spectrum, with a constant probability p_{null} . LL stands for log-likelihood, defined as the log-ratio of the probability of the match under the model, to that of the match under the null hypothesis.

For the details of computing log-likelihood score, please refer to Supplementary Text 4. We also added the following to the legend of Supplementary Table 18:

Log-likelihood probability for 14 different fragmentation models. $\#Max_L$ stands for the maximum number of bridge fragmentations allowed. $\#Max_C$ stands for the maximum number of the 2-cut fragmentations allowed. $\#Max_{2L+C}$ is an upper-bound on twice the number of 2-cuts plus the number of bridges (i.e. total number of cuts in the fragmentation). $Theo_{ef}$ stands for the size of the effective theoretical spectrum of a MSM, defined as peaks in the theoretical spectrum with annotated parents (the root is always annotated). Given a model θ , p_{θ} stand for the probability of a peak in the theoretical spectrum being explained by the model. The null hypothesis assumes peaks are explained independent of the theoretical spectrum, with a constant probability p_{null} . LL stands for log-likelihood, defined as the log-ratio of the probability of the match under the model, to that of the match under the null hypothesis.

C1.3. What is the correspondence of the pattern of fragmentation between substances of marine origin and those obtained from terrestrial origin, that is, what is the correspondence between the classes of marine and terrestrial biosynthesis?

R1.3. We looked at the frequency of different atoms and bonds in substances of marine and terrestrial origins. We modified Supplementary tables 16 and 17, adding the frequencies within marine and terrestrial classes. Our results show that these frequencies are similar. The caption of Tables S16 and S17 changes as follows:

Supplementary Table 16-17: $Freq_{marin}$ and $Freq_{terres}$ stand for frequency of different atoms/bonds within marine and terrestrial fractions of AntiMarin.

C1.4. Also, in cases where multiglycosylated or multitoxylated compounds such as glycosylated flavonoids are present, the depth number can easily exceed 3 levels.

R1.4. Our extensive analysis shows the reviewer is indeed correct (see our response R1.2). Now we are using the default depth $k=6$ for DEREPLICATOR+ webserver available through GNPS, and the depth parameter is also adjustable in the command-line version of the tool.

C1.5. Thus, I recommend the publication of the document, although the authors can provide a better explanation of the limitations found in the process of identification of secondary metabolites in the manuscript. I believe that the documentation on the in silico tools in the GNPS already presents some of the limitations of this dereplication tool - step 8 (<https://bix-lab.ucsd.edu/display/Public/Insilico+Peptidic+Natural+Products+Dereplicator+Documentation>).

R1.5. We thank the reviewer for pointing out this aspect. To clarify this point, we conducted a comprehensive analysis of “Spectra_{library}” which is the GNPS database of known natural products mass spectra. We added the following paragraph on the precision and recall rates of DEREPLICATOR+ on “Spectra_{library}” to the Supplementary Text 3.

Precision and recall rates. At 3% FDR threshold, DEREPLICATOR+ identifies 3087 spectra in *Spectra_{library}*, where for 1802 spectra the correct compound is ranked one, for 812 spectra the rank is two or three, and for 473 spectra the rank is four or above. If we define success as the prediction of the correct compound with rank 1, DEREPLICATOR+ has a recall rate of 32% (1802 out of 5473) and precision rate of 73% (1802 out of 3087). If we define success as the prediction of the correct compound among top 3 identifications, DEREPLICATOR+ has a recall rate of 47% (2614 out of 5473) and precision rate of 84% (2614 out of 3087).

To further investigate the limitations of DEREPLICATOR+, we evaluated the correlations between the success rate of DEREPLICATOR+ (in terms of assigning the correct compound as the top identification) and various properties of the compounds and spectra (compound mass, number of amide/NC/OC bonds, number of peaks in theoretical spectra, mass of spectra, number of peaks in experimental spectra). The only significant correlation that we observed was among the success rate and the number of NC-OC bonds in the structure. We added the following paragraph to the paper.

We further observed a positive correlation between the success rate of DEREPLICATOR+ in identifying the correct compound as the rank 1 MSM, and the total number of OC and NC bonds in the structure. For example, among compounds with 1-10 OC/NC bonds, 21% were correctly identified as the rank 1 MSMs, while the success rate increased to 38% for compounds with 11-20 OC/NC bonds, and 55% for compounds with 21-30 OC/NC bonds.

Further guidelines on running DEREPLICATOR+ are available through the following link, added to the caption of Supplementary Figure 1.

Guidelines and information on using DEREPLICATOR+ are available at : <https://bix-lab.ucsd.edu/display/Public/Insilico+Natural+Products+Dereplicator+Documentation>

Some specific suggestions and mistyping encountered in the manuscript.

C1.6. In “Development of chemical structure databases such as PubChem (...)”, the authors can include others appropriated compounds databases such as ChemBank, ChEMBL, ChEBI, DrugBank, NuBBEDB, Chemspider etc. (page: 1).

R1.6. We changed the following paragraph in the paper:

Development of chemical structure databases such as PubChem¹⁹ (≈83 million compounds), ChemSpider²⁰ (≈58 million compounds), ChEMBL²¹ (2.1 million compounds), ChemBank²² (1.2 million compounds), ChEBI²³ (440,000 compounds), Dictionary of Natural Products (≈300 thousand compounds), AntiMarin²⁴(≈60 thousand compounds), KEGG²⁵(≈16 thousand compounds), MetaCyc²⁶ (≈10 thousand compounds), mzCloud²⁷(≈3 thousand compounds), NuBBEDB²⁸ (2500 compounds), MIBiG²⁹(≈1600 compounds), DrugBank³⁰ (1360 compounds with structural information), and Norine³¹(≈1000 compounds) has paved the way for development of bioinformatics tools for natural product dereplication.

C1.7. In the text are encountered two forms of AntiMarin and Antimarin database, please choose one of them;

R1.7. We modified all occurrences of “Antimarin” to “AntiMarin”

C1.8. In “Analyzing Spectralibrary dataset” the word precursor is incorreced; Also in “Analyzing Spectralibrary dataset we suggest to replace reran by reanalyzed (page: 5)

R1.8. We fixed the typo in “precursor” and we replaced reran with reanalyzed

C1.9. In “Generating decoy fragmentation graphs” in methods, remove one of the “into” present in first line on page 9.

R1.9. We removed one of the “into”s.

C1.10. In “Annotating fragmentation graphs by spectra and scoring MSMs” also in methods sections correct the word spectrum (page 9).

R1.10. We fixed the typo in “spectrum”

C1.11. In addition, I suggest including an example for evaluation of the Dereplicator+ tool for peptide and a secondary metabolite on the GNPS platform.

R1.11. We added Supplementary Figure 1 as an example of running DEREPLICATOR+ on GNPS.

Figure 1 and Supplementary Figure 1 show the DEREPLICATOR+ pipeline that includes the following steps described in the Methods section.

We have also added the following link for the command-line version of DEREPLICATOR+. The command-line package of DEREPLICATOR+ now has sample spectra available for a peptide (Surugamide) and a polyketide (Chalcomycin).

URLs. DEREPLICATOR+ is available as both a stand-alone tool (<http://mohimanilab.cbd.cmu.edu/software/>) and a web application (<http://gnps.ucsd.edu/ProteoSAFe/static/gnps-theoretical.jsp>).

Further, we added the following description to the caption of Supplementary Figure 1.

Running DEREPLICATOR+ on the GNPS web server. (a) Image of the GNPS web page at www.gnps.edu. DEREPLICATOR+ users click on "Browse" in the section "GNPS Theoretical/Insilico" (b) At "DEREPLICATOR+, Identification of Metabolites Through Database Search of Mass Spectra" section, users click on "here". (c) To select input files, users click on "Select Input Files". Users can either import one of the existing GNPS public datasets from the "Share Files" section, or use their own data from "Upload Files" section. Users select a title/email for the job and specify the DEREPLICATOR+ accuracy mode (precursor and fragment ion mass tolerances). Users receive a notification email after the job is completed. (d) When the job is completed, users click on "View Significant Matches" to see all the identifications. Further guidelines and information on using DEREPLICATOR+ are available at: <https://bix-lab.ucsd.edu/display/Public/Insilico+Natural+Products+Dereplicator+Documentation>

We also added the links to the DEREPLICATOR+ search results through the GNPS web-server against *Spectra_{ActiSeq}* and *Spectra_{Lichen}* to the supplementary materials :

The search results for *Spectra_{ActiSeq}* and *Spectra_{Lichen}* are available at "<https://gnps.ucsd.edu/ProteoSAFe/status.jsp?task=5ea18d0c3b1d4018b7e0e79445ca8c18>" and "<https://gnps.ucsd.edu/ProteoSAFe/status.jsp?task=00c03bdbdae644e4935a8e094f249c2a>" respectively.

Reviewer #2 (Remarks to the Author):

The idea of natural products dereplication focuses on identifying already known compounds in a sample so further purification and bioassay experiments can focus on the uncharacterized compounds. The paper reports a minor improvement in an existing chemoinformatics approach, DEREPLICATOR which aims to computationally confirm the presence of known chemical structures in a biological sample by utilizing only mass spectral data. The parent approach is limited to peptides and authors have attempted to extend it to other chemical classes of natural products.

C2.1. While the paper tried to tackle a key challenge to identify non-peptide compounds in the mass spectral datasets, the proposed tool (DEREPLICATOR+) still need extensive testing and

validation in order to prove its versatility, accuracy and comprehensiveness to identify those compounds.

R2.1. We would like to thank the reviewer for the insightful comment on our paper. In response to the reviewer's comment, we highlighted the novel features in DEREPLICATOR+, in comparison to our previous tool DEREPLICATOR. We added the following sentence in "Analyzing *Spectral*_{library} dataset":

Overall, DEREPLICATOR+ identified 77% more compounds than DEREPLICATOR.

Furthermore, we added the following sentence to the "Discussion":

Our results show that the generalized bond breakage and the sequential fragmentation are crucial in enabling DEREPLICATOR+ to identify 77% more compounds (consisting of non-peptide metabolites and mixed peptide-polyketides) that were missed by DEREPLICATOR.

Overall, we believe that DEREPLICATOR+ is a significant improvement on previously published tools for identification of peptidic natural products that extended the functionality of DEREPLICATOR from peptides to other natural products. Indeed, increasing the identification rate by 77% as compared to existing tools would be considered a breakthrough rather than a minor improvement in traditional proteomics.

Major concerns are :

C2.2. Paper reports identification of very few chemicals from a small number of non-peptide chemical classes. This does not prove that the tool can reliably identify all the possible known chemicals of a non-peptide chemical class of natural products in a microbial sample's mass spectral dataset.

R2.2. DEREPLICATOR+ is not applicable to all classes of compounds. We recognize that it is a limitation but it is a limitation that applies to all existing metabolomics tools. DEREPLICATOR showed excellent performance on some compounds, e.g., it greatly improves on existing tools of peptidic natural products and represents the first tool able to identify the important class of peptides-polyketides and polyketides. Please see our response R2.24 below

C2.3. Identification of non-peptide compounds in mass spectral datasets is routinely performed using mass spectral library searches. Authors have not compared their tool's performance against these existing approaches and prove that the tool works better than that. It needs to be demonstrated that how many compounds the tool can identify which were missed by mass spectral library searches and vice versa.

R2.3. As the reviewer proposed, we benchmarked DEREPLICATOR+ against the spectral library search (see our responses R2.25 and R2.26 below). Spectral libraries indeed represent an excellent dereplication approach when the spectrum of a compound is known, which is not the case for many metabolites. Our benchmarking revealed that DEREPLICATOR+ is an excellent

tool for extending the currently limited spectral libraries.

C2.4. Tool's accuracy is not demonstrated to predict substructures for fragments in available known mass spectra for non-peptide compounds in public mass spectral libraries.

Only tool's specificity is shown but its sensitivity is also important for compound dereplication efforts. What is the confidence level that a compound suggested being absent by the tool is really not in the sample? A higher false negative rate can misguide investigators by not identifying compounds which were present in the sample and in the structural database. As the goal of the dereplication is to guide them on novel compounds, a higher false negative rate can direct them on something that is already present in structural databases. Data for both precision and recall need to be shown for the identification of non-peptide chemical classes.

R2.4. To be able to compute false negative rate, we need to run DEREPLICATOR+ on a dataset for which ground truth is known. We computed the precision and recall rates for "*Spectral_{library}*" dataset (for which the ground truth is known) and added the following paragraph to the Supplementary Text 3. DEREPLICATOR+ improved on MS-Finder with respect to the false negative rate.

Precision and recall rates. At 3% FDR threshold, DEREPLICATOR+ identifies 3087 spectra in *Spectral_{library}*, where for 1802 spectra the correct compound is ranked one, for 812 spectra the rank is two or three, and for 473 spectra the rank is four or above. If we define success as the prediction of the correct compound with rank 1, DEREPLICATOR+ has a recall rate of 32% (1802 out of 5473) and precision rate of 73% (1802 out of 3087). If we define success as the prediction of the correct compound among top 3 identifications, DEREPLICATOR+ has a recall rate of 47% (2614 out of 5473) and precision rate of 84% (2614 out of 3087).

C2.5. Isomeric compounds are present in mass spectral datasets for a biological specimen. Paper did not explain how the tool discriminates those compounds.

R2.5. The results in our spectral library search section shows that our tool successfully distinguishes isomeric compounds. We added two columns to our Supplementary Table 7 showing the chemical formula and number of isomeric compounds in the DNP database. We added the following paragraph to the paper.

Isomer compounds. Compounds in *Spectral_{library}* have on average 8 isomers with identical chemical formula in DNP. Among 4360 compounds (80%) of *Spectral_{library}*, which have at least one other isomer in the DNP database, DEREPLICATOR+ correctly identified 1746 (40%) of compounds (Supplementary Table 7).

C2.6. During the electrospray ionization, a compound can generate multiple adducts and in-source fragments which can have different MS/MS spectra (<https://pubs.acs.org/doi/10.1021/acs.analchem.7b02380> and <https://www.ncbi.nlm.nih.gov/pmc/articles/PMC5750104/>). How the tool handles these mass spectra is not explained. Theoretical spectrum of these adducts and fragments were not generated by the tool. Were they identified as different compounds?

R2.6. We added the capability to search for adducts such as sodium or potassium adducts for DEREPLICATOR+, and added the following paragraph to the “Spectral library analysis section”

Adducts. To assess the capability of DEREPLICATOR+ in identifying adducts, we searched 1207 annotated spectra with sodium/potassium adducts from the GNPS spectral library using DEREPLICATOR+, and 280 (23%) of them were correctly identified at 1% FDR, while 23 (2%) were falsely identifies as non-adduct compounds (Supplementary Table 9).

C2.7. Up to 95% natural products in the DNP database are below 1000Da. However the tool seems to be working better for compounds above this threshold. Tool’s performance for compounds below 1000Da needs to be shown thoroughly.

R2.7. We have evaluated the performance of the tool for various mass ranges - see our response R2.33 below. We view the good performance of DEREPLICATOR+ on large masses as a plus rather than minus since it complements some tools that are limited to small masses.

Below are specific comments.

Title :

C2.8. “natural products” should be changed to “microbial natural products” as the paper does not have data for plant-specific natural products.

- Replace “in-silico identification” with “dereplication”. The word “Identification” gives the impression that it can identify completely unknown compounds in a sample which is not correct. The tool starts with a compound structure database that will be searched against a mass spectral dataset to confirm only the presence of those compounds.

R2.8. According to the reviewer suggestion, we changed the title to

“Dereplication of Microbial Metabolites Through Database Search of Mass Spectra”

Abstract :

C2.9. “other classes” should be replaced with exact chemical classes that the tool can identify. The main compelling idea of the paper is that the tool can go beyond peptides and lipids, I think it is important to show those classes in abstract.

R2.9. The main idea of the paper is that DEREPLICATOR+ improves on peptide identification as compared to DEREPLICATOR and goes beyond peptides (DEREPLICATOR does not work for lipids). We modified the abstract as follows :

We developed a DEREPLICATOR+ algorithm that improved on previous approaches for identifying peptidic natural products and extended them for identifying polyketides, terpenes, benzenoids, alkaloids, flavonoids, and other classes of natural products.

Introduction :

Misleading statements :

C2.10. “most of the existing tools for metabolite identifications work best for identification of small molecule (below 500 Da) and become prohibitively time-consuming for searching large spectral datasets”. LipidBlast can identify lipids up to 2000 Da (<https://www.nature.com/articles/nmeth.2551>). And, NIST MS Search software (<http://chemdata.nist.gov/dokuwiki/doku.php?id=peptidew:mspepsearch>) can search against large spectral datasets.

R2.10. We modified the following paragraph

Currently, the fast spectral library search programs⁵⁰ searches over 1000 spectra against the entire NIST library per seconds. However these approaches are unable to search the chemical structure libraries. Despite recent progress (CSI:FingerID⁴² increased metabolite identification rates five-fold as compared to previous approaches), the existing tools for metabolite identifications are either limited to a specific class of molecules such as peptides and lipids^{45,46}, work best for identification of small molecules (below 500 Da)⁴², or become prohibitively time-consuming for searching large spectral datasets⁴².

C2.11. “approaches that construct in silico mass spectra by fragmenting natural peptides along specific bonds^{39,40}”. LipidBlast (ref number 40) generates in-silico spectra for lipids not peptides.

R2.11. We modified the following line :

approaches that construct in silico mass spectra by fragmenting peptide and lipid natural products along specific bonds^{45,46}.

Results

Experimental datasets (MSV000078604 , MSV000078839, MSV000078568, MSV000078584)

C2.12. Provide ionization mode, liquid chromatography parameters and mass spectrometry instrument details for the experimental datasets.

R2.12. We added the following as Supplementary Text 1. Datasets :

MSV000078839 dataset. 36 strains of *Streptomyces* were grown on A1, MS and R5 agar, extracted sequentially with ethyl acetate, butanol and methanol, and analyzed on Agilent 6530 Accurate-Mass Q-TOF spectrometer coupled to a C18 RP Agilent 1260 LC system (ESI ionization, CID fragmentation).

MSV000078604 dataset. 16 strains of *Streptomyces* were grown on ISP2 agar plates, extracted by butanol, and analyzed on LTQ Orbitrap Velos coupled to a C18 RP Agilent 1200 LC system (ESI ionization, HCD fragmentation).

MSV000078568 dataset. A total of 317 cyanobacterial collections were extracted repetitively with CH₂Cl₂:MeOH 2:1, dried in vacuo, and fractionated into nine fractions (A-I) by silica gel vacuum liquid chromatography (VLC) using a stepwise gradient of hexanes/EtOAc and

EtOAc/MeOH, and analyzed on a Maxis Impact mass spectrometer coupled to C18 RP-UHPLC (ESI ionization, CID fragmentation).

MSV000078584 dataset. Metabolites from 110 spots of lichen were extracted with 4:1 ethyl acetate-methanol and 0.1% trifluoroacetic acid (TFA), and analyzed on Maxis Q-TOF mass spectrometer (Bruker Daltonics) coupled to a C18 RP UltiMate 3000 UHPLC (ESI ionization, CID fragmentation).

C2.13. Ionization modes gives different mass spectra for the same compound. How the tool perform for both ESI (-) and ESI (+) modes ?

R2.13. DEREPLICATOR+ can handle spectra with negative charges. Currently, all large-scale GNPS datasets are collected on ESI (+) mode. However, the GNPS spectral library has 341 compounds in ESI (-) mode. We ran DEREPLICATOR+ on these 341 compounds, and added the following paragraph:

Negatively ionized spectra. To assess the capability of DEREPLICATOR+ in identifying compounds in the negative ionization modes, we analyzed 341 additional spectra in the GNPS spectral library collected in the ESI (-) mode. DEREPLICATOR+ search correctly identified 88 (26%) of these compounds as top predictions (Supplementary Table 10).

C2.14. The quality of mass spectra depends on the utilized instrument. How the tool performs for data acquired on different instruments ?

R2.14. We added the following paragraph to the manuscripts.

Comparison between different instruments. Among GNPS datasets (317.2 million spectra) in *Spectra_{GNPS}*, 72.7% (230.1 million) are from the TOF instruments, while 18.1% (57.5) million are from the Orbitrap instruments. Among identifications in *Spectra_{GNPS}*, 4.4 million (62.0%) are from the TOF instruments and 2.3 million (32.8%) are from the Orbitrap instruments (Supplementary Table 15).

C2.15. HCD or CID spectra are different for several compounds. Was the tool tested that it works for both type of spectra with same accuracy?

R2.15. We modified Supplementary Table 2 and added a new column that shows whether the compound was discovered in CID or HCD spectral dataset. We added the following paragraph to the manuscript :

Comparison of CID and HCD spectra. We divided *Spectra_{ActiSeq}* into two parts, *Spectra_{ActiSeq-CID}* consisting of 473135 spectra from MSV000078839 dataset and *Spectra_{ActiSeq-HCD}* consisting of 178635 spectra from MSV000078604 dataset. DEREPLICATOR+ identified 2979 spectra (129 compounds) in *Spectra_{ActiSeq-CID}* and 5215 spectra (404 compounds) in *Spectra_{ActiSeq-HCD}*. 45 compounds are found in both CID and HCD datasets. Supplementary Table 2 shows compounds discovered in CID and HCD spectra at 1% FDR.

C2.16. How it was ensured that MS/MS spectra in these experimental datasets were clean and did not contain any isobaric species or contaminant fragments (<https://www.ncbi.nlm.nih.gov/pubmed/18790129>) ? Were any quality checks and filtering were applied on the spectra before matching them against the known structures?

R2.16. We used a filtering similar to that used in Molecular Networking and MS-Cluster tools. We added the following paragraph to the paper:

Similar to MS-Cluster and Molecular Networking approaches^{56,57}, six top intensity peaks are retained in each window of size 50 Da within each spectrum.

While we did not perform any quality control, DEREPLICATOR+ did not incorrectly identify any of these contaminants as a natural product at 1% FDR. We added the following paragraph to the paper :

Mass spectrometry contaminants. To evaluate whether DEREPLICATOR+ incorrectly identifies common mass spectrometry contaminants as natural products, we performed an evaluation of the masses of 760 contaminants from Keller *et al.* . Among these 760 masses, 319 are present at 0.02 Da threshold in *SpectraActiSeq*. However, only m/z 1020.50 (bovine trypsin) identified by DEREPLICATOR+ and reported as Salinamide A at 1% FDR (Supplementary Table 4). We manually confirmed that spectra at m/z 1020.50 is indeed Salinamide A and not a contaminant (Supplementary Figure 4).

Structural databases (DNP and AntiMarin):

C2.17. DNP database has almost 150,000 compounds. Why only 83889 were used ? Was any filtering on the database was applied ? Tool need to recalculate the results for entire DNP database.

R2.17. As of August 2017, the total number of compounds in DNP is 254727. However, only 83889 of such compounds are unique. As an example, the DNP has two distinct entries: “Fusarinine_C_N2,N14,N26-tri-Ac” and “Fusarinine_C_N2,N14,N26-Tri-Ac”. These two entries have identical chemical structures, but distinct identifiers in DNP. To better estimate the number of unique identified compounds, we remove all the redundant entries from the DNP and AntiMarin datasets. We modified the following paragraph to explain this pre-processing.

Datasets. To benchmark DEREPLICATOR+, we used the AntiMarin database (60908 compounds, 29491 unique compounds after removing duplicates) and the Dictionary of Natural Products (254727 compounds, 83889 unique compounds after removing duplicates) to dereplicate all spectra from the following spectral data sets specified in Supplementary Table 1 and Supplementary Text 1. Compounds are flagged as duplicates if they have identical chemical structures.

C2.18. DNP has 99 compounds listed for Actinomyces species. Were these compound present in the DNP-subset used in the paper.

R2.18. Please see our comment R.27 below

C2.19. How tool perform for entire PubChem database? Can it identify more compounds in the experimental mass spectra datasets using PubChem? DNP is a small subset database of possible natural products. Searching against PubChem shall identify many more compounds in these mass spectral datasets.

R2.19. We added the following paragraph.

DEREPLICATOR+ search of *SpectraActiSeq* against PubChem, AntiMarin, HMDB, LipidMaps, DNP, DrugBank, GNPS spectral library, KEGG, MiBIG, StreptomeDB, and UNPD, identified 539 compounds at 1% FDR.

C2.20. “Spectralibrary “ has associated chemical structures. The tool needs to generate spectra for these structures and show that how many were accurate.

R2.20. This experiment has been performed. Please note that in addition to the DNP, we input all the 5473 chemical structures from the *SpectraLibrary* for the search, and DEREPLICATOR+ automatically generates the theoretical spectra from the input chemical structures. To clarify this point, we modified the first paragraph in the section “Analyzing *SpectraLibrary* dataset” as follows:

To benchmark the accuracy of DEREPLICATOR+ in identification of spectra from known compounds, we searched 5,473 annotated spectra from *SpectraLibrary* against a database of their 5,473 chemical structures, plus 83,889 unique chemical structures from the DNP database. Then we removed 2697 duplicate compounds that were shared between the spectral library and DNP database, resulting in 86,665 unique compounds.

Analyzing SpectraActiSeq dataset

C2.21. “66 unique compounds (148 MSMs)” indicate that 148 spectra belonging to these 66 unique compounds. Count of identified compound increased by two-fold however the count of identified spectra increased by 18 fold, why ?

R2.21. We added the following paragraph to the paper :

DEREPLICATOR+ not only identified more unique compounds at the same FDR threshold, but also identified more spectra per compounds (average of 2.2 spectra per compound for DEREPLICATOR, versus 16.7 spectra per compound for DEREPLICATOR+). This is partially because spectra from the same compound often differ in the quality of fragmentation. DEREPLICATOR is mainly limited to identification of the highest quality spectra since it uses a rather restrictive fragmentation model. DEREPLICATOR+ identifies spectra of lower quality since it uses a more detailed fragmentation model.

C2.22. It should be stated that only 1.5% (2666/178635) of all spectra were identified.

R2.22. We added the following sentence to the paper :

At this FDR threshold, DEREPLICATOR+ identified 1.5% of the spectra in *SpectraActiSeq* dataset.

C2.23. DNP provides the class information for compounds. ClassyFire poorly covers many natural product classes that DNP has. What were the classes for the identified compounds in the DNP database ?

R2.23. We have now computed classes for all identifications from *SpectraGNPS*. Please see our comment R2.31 below

C2.24. Statement “Table 1 describes 25=29-4 compounds (covering 20 PNP, 2 polyketides, 2 terpenes, and 1 benzenoid) identified by DEREPLICATOR+.” indicates that the tool still needs improvement to comprehensively cover non-peptide chemical classes. Just identifying one compound from a class does not prove that it should be able to identify other chemicals in that class.

R2.24. We agree with the reviewer that identifying a single compound from a class does not prove that it should be able to identify other chemicals in that class. However, *SpectraActiSeq* is a very small sub-dataset of *SpectraGNPS*. To address this comment, we modified the following paragraph :

For DEREPLICATOR+ identifications, 2913 (54.6%) were classified as peptides, 1595 (29.9%) as lipids, 272 (5.1%) as organic oxygen compounds, 160 (3.0%) as organoheterocyclic compounds, 155 (2.9%) as polyketides and 133 (2.5%) as benzenoids.

Moreover, we added the following sentence to the caption of Figure 2:

Over 33% (155 out of 464) of DEREPLICATOR+ identifications have a single or no amide bonds.

Moreover, we added the following sentence to the discussion section:

Our results show that generalized bond breakage and the sequential fragmentation are both crucial in enabling DEREPLICATOR+ to identify 77% more compounds (consisting of non-peptide metabolites and mixed peptide-polyketides) that were missed by DEREPLICATOR. Moreover, over 33% of compounds identified by DEREPLICATOR+ have either a single or no amide bond.

C2.25. Authors have compared DEREPLICATOR+ against its parent tool. But it must be demonstrated that the tool works better than existing mass spectral matching approaches to identify non-peptide compounds in experimental mass spectral datasets. Specifically, the overlap among compounds identified using NIST17, LipidBlast and MONA databases and DEREPLICATOR+ need to be computed.

R2.25. We searched SpectraActiSeq against NIST17, LipidBlast and MoNA spectral libraries, and added the following subsection.

Benchmarking against the spectral library search. The NIST spectral library search toolkit MSPepSearch identified 34, 12, and 12 compounds in searching SpectraActiSeq against NIST17, LipidBlast and MoNA. At 1% FDR, DEREPLICATOR+ recovered 27/34 identifications from NIST, 8/12 identifications from LipidBlast, and 9/12 identifications from MoNA (Supplementary Table 6). At this FDR threshold DEREPLICATOR+ identified 315 compounds, 272 of them absent from LipidBlast, MoNA and NIST search results.

C2.26. The tool also need to be compared with the MS-FINDER software (<https://pubs.acs.org/doi/abs/10.1021/acs.analchem.6b00770>) which can also predict substructures for product ions.

R2.26. We searched the *Spectral_{library}* using MS-FINDER and added the following paragraph :

MS-FINDER⁴¹ identified 20% of compounds from *Spectral_{library}* as top compounds, 27% as top three, and 38% as top 10 compounds using a precursor and product ion tolerance of 0.02Da similar to what we used for DEREPLICATOR+ (Supplementary Table 8).

C2.27. DNP lists 99 compounds for Actinomyces species. How many of them were missed by the tool ?

R2.27. If DEREPLICATOR+ does not identify a compound in *Spectra_{ActiSeq}* it does not necessarily mean that the tool missed the compound. It could also mean either (i) the Actinomyces producer was not among the strains analyzed in the study, and (ii) the strain is not producing the compound under the specific conditions of the study. We added the following paragraph to the manuscript.

Among 207 compounds from DNP annotated as “Actinomyces/Streptomyces”, DEREPLICATOR+ identified 56 of them in *Spectra_{ActiSeq}* (Supplementary Table 3).

Analyzing Spectralibrary dataset.

C2.28. “This reduced the search space from 135 candidate structures to only 24 candidate structures per spectrum, and increased the number of identifications on the top to 44%(2414), 40%(2216) and 25%(1362) for DEREPLICATOR+, DEREPLICATOR-G and DEREPLICATOR, respectively.” What classes were covered among these 2414 identified compounds ? How many of them were non-peptides ? Why 56% spectra could not be identified by the tool ?

R2.28. While DEREPLICATOR+ identifies only 44% of the compounds at rank 1, even in cases in which it can not accurately put the correct compound at rank 1, it assigns a high score and low rank to the correct compound. We added the following sentence to the paper.

Furthermore, for 62.8% (3441) and 89.3% (4890) of the cases, the correct compound was identified within top three and top ten ranked compounds by DEREPLICATOR+.

We also looked at the classes of identified compounds in DNP. We added a new column to Supplementary Table 7 specifying DNP classification, and the following sentence:

Among DEREPLICATOR+ identification of spectra from *SpectraLibrary* with DNP classification available (224 compounds), 28% (63) are classified by DNP as terpenes, 14% (32) are alkaloids, 13% (29) as flavonoids, 13% (28) as peptides, 8% (18) as aliphatic natural products, 8% (18) as simple aromatic natural products, 5% (11) as benzopyranoids, 4% (9) as steroids, 2% (5) as carbohydrates, 2% (5) as oxygen heterocycles, 1% (2) as lignans, 1% (2) as polycyclic aromatic natural products, 0.5% (1) as polyketides, and 0.5% (1) as benzofuranoids (Supplementary Table 7).

Analyzing SpectraLichen dataset and Analyzing SpectraCyan dataset.

C2.29. Which DNP classes were covered ? ClassyFire is not a relevant classification ontology for natural products.

R2.29. We now have computed classes for the entire GNPS. Please see our comment R2.31 below

Assigning almiramide biosynthetic gene cluster.

C2.30. It needs to be shown that the parent DEREPLICATOR tool failed to identify this compound and only DEREPLICATOR+ was able to do so. This compound is still a peptide. Did author identify any gene cluster for non-peptide natural products ?

R2.30. Alimiramide is a NRPS-PKS mix, and DEREPLICATOR failed to identify this compound at 1% FDR threshold. We added the following sentence to the paper.

At 1% FDR threshold, DEREPLICATOR failed to identify this NRPS-PKS.

Analyzing SpectraGNPS dataset.

C2.31. “For DEREPLICATOR+ identifications, 54.6% were classified as peptides, 29.9% as lipids, 5.1% as organic oxygen compounds, 3.0% as organoheterocyclic compounds, 2.9% as polyketides and 2.5% as benzenoids.” . Which DNP classes were covered in this ? Lipids structures can be dereplicated using the LipidBlast library so the lipid identification need to be compared with results from LipidBlast annotation.

R2.31. We added the following paragraph to the paper :

Among 5,336 DEREPLICATOR+ identifications in *Spectra_{GNPS}*, 692 have class annotations in the DNP database. Among them, 227 (32.8%) are terpenoids, 150 (21.6%) are aliphatic natural products, 101 (14.5%) are alkaloids, 69 (9.9%) are peptides, 39 (5.6%) are simple aromatic

natural products, 31 (4.4%) are steroids, 21 (3.0%) are flavonoids, 14 (2.0%) are benzopyranoids, 12 (1.7%) are lignans, 7 (1.0%) are oxygen heterocycles, 7 (1.0%) are oxygen heterocycles, 5 (0.7%) are carbohydrates, 5 (0.7%) are polypyrroles, and 3 (0.5%) are polyketides.

For comparison to LipidBlast, please see our response R2.25

C2.32. Results on peptides can be shortened as the main point of the paper is to identify non-peptide chemical classes.

R2.32. The key point of this paper is a new approach to analyzing metabolite fragmentation and scoring Metabolite-Spectrum Matches. Since applications of this new approach cover both peptides and other natural products, we believe that identification of peptides and non-peptides are two equally important contributions. Moreover, an automated identification of peptide-polyketides and other peptide-mix compounds is a crucial open problem in the field of natural products discovery that has not been properly addressed before.

Previous software tools such as DEREPLICATOR and MS-FINDER represented initial steps toward solving this open problem in the field, and improved models are necessary for a comprehensive solution to this problem. Our results show that DEREPLICATOR+ greatly improves on DEREPLICATOR in identification of peptide-mix compounds. Therefore, we decided to not remove or shorten the paragraph describing the advantage of DEREPLICATOR+ over DEREPLICATOR in identification of peptide-mixes. We added the following paragraph clarifying the issue.

DEREPLICATOR+ identified 32% more peptide natural products, and overall 77% more compounds than DEREPLICATOR at 0.02Da tolerance.

Discussion

C2.33. Statement “While many of the existing approaches focus on small metabolites with mass below 1000Da³⁶, DEREPLICATOR+ works for both small and large metabolites.” -is overstated and can be toned down.

- LipidBlast can identify lipids up to 2000Da.
- Molecular weights distribution of the identified compounds by dereplicator+ is not provided in the paper to support this claim. DNP database says that only 5% of its compounds are above 1000Da. It seems that a majority of natural products are below 1000 Da but the majority of the identified compounds by the tool are above 1000 Da which is a small fraction of the natural products.

R2.33. We removed the sentence :

While most of the existing approaches focus on small metabolites with mass below 1000Da³⁶, DEREPLICATOR+ works for both small and large metabolites.

Our analysis showed that the majority (3802, 71%) of 5336 compounds identified in GNPS have masses below 1000Da. We added the following paragraph to the paper.

Among these 5,336 identifications, 643 (12%) have mass below 400, 2421 (45%) have masses between 400 to 800, 1425 (26%) have masses between 800Da to 1200Da, and 847 (15%) have masses above 1200Da. Supplementary Figure 6 shows the distribution of masses of compounds identified by DEREPLICATOR+ in *Spectra_{GNPS}*.

REVIEWERS' COMMENTS:

Reviewer #1 (Remarks to the Author):

The reading of the authors answers letter showed the concern in answering all the comments of my reviewer.

The analysis of the submitted file confirms the great advance in the quality of the manuscript.

In our previous analysis we highlight the novelty, the interest by other communities. After the corrections there are no technical problems and I'm very convince that this article will have a great impact.

In summary, I believe that it can now be publish at the present form.

Reviewer #2 (Remarks to the Author):

Authors have made significant changes in the manuscript and have satisfied most of my comments on it.

I have few additional minor comments.

The issue of false-negative rate can be highlighted in the main text. It shall also be mentioned in the main text that LCMS instruments were used for acquiring the tested datasets.

Comments on replies :-

R2.1 - DEPLICATOR+ failed to correctly identify 66% (3595) spectra in the SpectraLibrary dataset even though true positive structures were present in the searched structure database. This is the false-negative error rate for the tool and it should be mentioned in the main text in the "Analyzing SpectraLibrary dataset" section. Authors should discuss what future directions can be adopted to improve this error rate.

R2.2 - Add the reply "DEREPLICATOR+ is not applicable to all classes of compounds. We recognize that it is a limitation but it is a limitation that applies to all existing metabolomics tools. DEREPLICATOR showed excellent performance on some compounds, e.g., it greatly improves on existing tools of peptidic natural products and represents the first tool able to identify the important class of peptides-polyketides and polyketides." to the main text.

R 2.4 - Add the reply to the main text.

R 2.12 – satisfied. But, mention "reversed-phase liquid chromatography high resolution mass spectrometry" in the main text.

REVIEWERS' COMMENTS:

Reviewer #1 (Remarks to the Author):

The reading of the authors answers letter showed the concern in answering all the comments of my reviewer.

The analysis of the submitted file confirms the great advance in the quality of the manuscript. In our previous analysis we highlight the novelty, the interest by other communities. After the corrections there are no technical problems and I'm very convince that this article will have a great impact.

In summary, I believe that it can now be publish at the present form.

Our response. We thank the reviewer for helping us in improving the manuscript.

Reviewer #2 (Remarks to the Author):

Authors have made significant changes in the manuscript and have satisfied most of my comments on it.

I have few additional minor comments.

The issue of false-negative rate can be highlighted in the main text. It shall also be mentioned in the main text that LCMS instruments were used for acquiring the tested datasets.

Comments on replies :-

R2.1 - DEPLICATOR+ failed to correctly identify 66% (3595) spectra in the SpectralLibrary dataset even though true positive structures were present in the searched structure database. This is the false-negative error rate for the tool and it should be mentioned in the main text in the "Analyzing SpectralLibrary dataset" section. Authors should discuss what future directions can be adopted to improve this error rate.

Our response. We modified and moved the following paragraph from supplementary to the main text

Precision and recall rates. At 3% FDR threshold, DEREPLICATOR+ identifies 3087 spectra in *Spectra_{library}*, where for 1802 spectra the correct compound is ranked one, for 812 spectra the rank is two or three, and for 473 spectra the rank is four or above. If we define success as the prediction of the correct compound with rank 1, DEREPLICATOR+ has a recall rate of 33% (1802 out of 5473) and precision rate of 58% (1802 out of 3087). If we define success as the prediction of the correct compound among top 3 identifications, DEREPLICATOR+ has a recall rate of 48% (2614 out of 5473) and precision rate of 85% (2614 out of 3087). DEPLICATOR+ failed to correctly identify 67% (3671 out of 5473) spectra in the *Spectral_{Library}* dataset even though true positive structures were present in the database.

Oure response. We also modified the following paragraph in the discussions :

Currently DEREPLICATOR+ uses a simple shared peaks scoring to match spectra against fragmentation graphs, and computing false discovery rate for the found identifications. **This simple scoring scheme results in a high false negative rate.** Meanwhile, DEREPLICATOR+ enables automatic collection of large metabolite spectra libraries by searching billions of spectra, similar to those in proteomics. **This pave the path for machine learning methods to improve scoring scheme and enhance the false negative rates.**

R2.2 - Add the reply “DEREPLICATOR+ is not applicable to all classes of compounds. We recognize that it is a limitation but it is a limitation that applies to all existing metabolomics tools. DEREPLICATOR showed excellent performance on some compounds, e.g., it greatly improves on existing tools of peptidic natural products and represents the first tool able to identify the important class of peptides-polyketides and polyketides.” to the main text.

Our response. We added the following to the discussion section.

Similar to all the existing metabolomics tools, DEREPLICATOR+ is not applicable to all the classes of compounds. DEREPLICATOR+ greatly improves on the existing tools for peptidic natural products and represents the first tool able to identify the important class of peptides-polyketides and polyketides.

R 2.4 - Add the reply to the main text.

Oure response. We moved the following paragraph from supplementary to the main text

Precision and recall rates. At 3% FDR threshold, DEREPLICATOR+ identifies 3087 spectra in *Spectra_{library}*, where for 1802 spectra the correct compound is ranked one, for 812 spectra the rank is two or three, and for 473 spectra the rank is four or above. If we define success as the prediction of the correct compound with rank 1, DEREPLICATOR+ has a recall rate of 32% (1802 out of 5473) and precision rate of 73% (1802 out of 3087). If we define success as the prediction of the correct compound among top 3 identifications, DEREPLICATOR+ has a recall rate of 47% (2614 out of 5473) and precision rate of 84% (2614 out of 3087). DEREPLICATOR+ failed to correctly identify 66% (3671 out of 5473) spectra in the *Spectral_{Library}* dataset even though true positive structures were present in the database.

R 2.12 – satisfied. But, mention “reversed-phase liquid chromatography high resolution mass spectrometry” in the main text.

Our response. We modified following sentence in the main text :

We searched all spectra from the following **reversed-phase liquid chromatography high resolution mass spectrometry** datasets specified in Supplementary Table 1 and Supplementary Note 1.